# Revealing protonation states and tracking substrate in serine hydroxymethyltransferase with room-temperature X-ray and neutron crystallography

Victoria N. Drago[1], Claudia Campos[2], Mattea Hooper[2], Aliyah Collins[2], Oksana Gerlits[2], Kevin L. Weiss [1],
Matthew P. Blakeley [3], Robert S. Phillips [4,5] & Andrey Kovalevsky [1✉]

Pyridoxal 5′-phosphate (PLP)-dependent enzymes utilize a vitamin B$_6$-derived cofactor to perform a myriad of chemical transformations on amino acids and other small molecules. Some PLP-dependent enzymes, such as serine hydroxymethyltransferase (SHMT), are promising drug targets for the design of small-molecule antimicrobials and anticancer therapeutics, while others have been used to synthesize pharmaceutical building blocks. Understanding PLP-dependent catalysis and the reaction specificity is crucial to advance structure-assisted drug design and enzyme engineering. Here we report the direct determination of the protonation states in the active site of *Thermus thermophilus* SHMT (*Tth*SHMT) in the internal aldimine state using room-temperature joint X-ray/neutron crystallography. Conserved active site architecture of the model enzyme *Tth*SHMT and of human mitochondrial SHMT (hSHMT2) were compared by obtaining a room-temperature X-ray structure of hSHMT2, suggesting identical protonation states in the human enzyme. The amino acid substrate serine pathway through the *Tth*SHMT active site cavity was tracked, revealing the peripheral and cationic binding sites that correspond to the *pre*-Michaelis and *pseudo*-Michaelis complexes, respectively. At the peripheral binding site, the substrate is bound in the zwitterionic form. By analyzing the observed protonation states, Glu53, but not His residues, is proposed as the general base catalyst, orchestrating the retro-aldol transformation of L-serine into glycine.

[1] Neutron Scattering Division, Oak Ridge National Laboratory, Oak Ridge, TN 37831, USA. [2] Department of Natural Sciences, Tennessee Wesleyan University, Athens, TN 37303, USA. [3] Large Scale Structures Group, Institut Laue–Langevin, 71 Avenue des Martyrs, 38000 Grenoble, France. [4] Department of Chemistry, University of Georgia, Athens, GA 30602, USA. [5] Department of Biochemistry and Molecular Biology, University of Georgia, Athens, GA 30602, USA. ✉email: kovalevskyay@ornl.gov

Found in all living organisms, pyridoxal 5'-phosphate (PLP)-dependent enzymes utilize a phosphorylated, biologically active form of vitamin $B_6$ (Fig. 1a) to catalyze a myriad of chemical reactions including transamination, β- and γ-elimination, α-decarboxylation, retro-aldol cleavages, and glycogen phosphorylation[1,2]. PLP-dependent enzymes have been organized into seven fold types based on their evolutionary lineages, however, within these groupings, there are multiple catalytic activities[3–5]. The largest group of PLP-dependent enzymes is the aminotransferase superfamily (Fold I). Hence, PLP-dependent enzymes, owing to their versatility, are physiologically significant for the metabolism, interconversion, and synthesis of various amino acids. Unsurprisingly, many PLP-dependent enzymes are recognized as drug targets for the development of small-molecule therapeutics and have also been exploited for the synthesis of building blocks for pharmaceutical applications[6,7].

PLP-dependent catalysis is enabled by the stereoelectronic control of the labile covalent linkage, called the Schiff base, formed between the PLP cofactor and the side-chain amino group of a catalytic lysine (Lys) residue to create the internal aldimine functional state (Fig. 1a), and by the ability of the enzyme active sites to stabilize charged intermediates through the electron withdrawing properties of the pyridine ring[8–10]. These principles, however, do not fully explain the catalytic diversity of the PLP-dependent enzymes, but rather it appears that the manifold of PLP-dependent activities is governed by the surrounding protein environment and the electronic modulation of the cofactor through selective protonation. Consequently, PLP-dependent catalysis cannot be completely understood without knowing the protonation states of the cofactor and the surrounding residues that are dictated by the locations of hydrogen (H) atoms, which determine the electrostatic environment within the active site.

Serine hydroxymethyltransferase (SHMT), a ubiquitous PLP-dependent enzyme in the aminotransferase superfamily, catalyzes the reversible conversion of L-serine (L-Ser) to glycine (Gly), transferring a one-carbon unit to tetrahydrofolate (THF) to yield 5,10-methylenetetrahydrofolate (5,10-MTHF) (Fig. 1b)[5,11]. SHMT exhibits some catalytic promiscuity, similar to many PLP-dependent enzymes, and can catalyze THF-independent reactions such as the reversible cleavage of β-hydroxy amino acids[12], decarboxylation of aminomalonates[13], and the racemization and transamination of D- and L-alanine[14]. As an enzyme in one-carbon metabolism, SHMT is essential to the synthesis of thymine nucleotides, purines, methionine, and other essential biomolecules[15,16]. SHMT is found as a homodimer in prokaryotes, such as *Thermus thermophilus* (*Tth*), and a homotetramer in eukaryotes, such as humans (Fig. 1c and d)[17,18]. One-carbon metabolism is compartmentalized in mammals and hence requires two SHMT isoforms, SHMT1 and SHMT2, located in the cytosol and mitochondria, respectively[18,19]. SHMT1 and SHMT2 catalyze equivalent biochemical reactions and have a sequence identity of ~66%[20]. Interestingly, in many cancer cell lines, these reactions proceed in opposite directions in SHMT1 and SHMT2, with SHMT2 providing the majority of one-carbon units in mitochondria from serine synthesized by SHMT1 in the cytosol[21].

Human SHMT2 (hSHMT2, Fig. 1c) has been found to be highly overexpressed in many types of cancers as a result of metabolic reprogramming of the one-carbon metabolism[22–27]. Additionally, high levels of hSHMT2 expression have been linked to poor cancer prognoses and to resistance to 5-fluorouracil, one of the most commonly used drugs to treat cancer[28–31]. The significance of hSHMT2 in the proliferation and drug resistance of cancer cells has made this enzyme an attractive drug target for antimetabolite chemotherapies[16,32,33], and several promising hSHMT2 inhibitors have been discovered[34–37]. To advance drug design[38] of hSHMT2-specific inhibitors, it is critical to understand the SHMT catalysis at the atomic level. However, there is a gap in our knowledge of the SHMT-catalyzed reaction mechanism, which is based on acid-base chemistry, because the locations and movement of H atoms along the reaction pathway are not known.

Hydrogen (H) atoms comprise ~50% of all atoms in proteins and contribute to most noncovalent interactions, such as H-bonding, van der Waal's forces, and electrostatics. Thus, H atoms are key players in enzyme substrate and ligand binding, protein-protein interactions, and proton transfer reactions[39,40]. Moreover, the presence or absence of an H atom in a chemical group, such as imidazole of histidine (His) or carboxylate of aspartate/glutamate (Asp/Glu), determines the protonation state and thereby the electrical charge of that amino acid residue. Whereas X-ray diffraction continues to be the gold standard in protein structure determination, the resultant structures provide little to no information on the positions of functional H atoms even at ultrahigh resolutions[41]. This is a consequence of the X-ray scattering magnitude of H being significantly smaller than that for the other atoms in biomacromolecular structures. Because H and its heavier isotope deuterium (D) atoms have comparable neutron scattering lengths to heavier atoms (C, N, O) within the protein, neutron diffraction can resolve the positions of H/D at moderate resolutions[42–45]. The ability to assign protonation states based on the location of H/D atoms by neutron diffraction is critical to understanding the mechanism of catalysis, molecular recognition, and in designing specific inhibitors through structure-based drug design[46–50]. As a result, a neutron structure can help resolve a biochemical issue for a specific protein[51,52] and provide an accurate model for drug design[49,50].

The first step in the SHMT-catalyzed conversion of L-Ser to Gly is the formation of the external aldimine through a trans-aldimination reaction with the L-Ser substrate, a common feature for PLP-dependent enzymes. The amino group of L-Ser displaces the catalytic Lys, initially covalently linked to PLP within the internal aldimine state, to generate an external aldimine state followed by THF binding that ultimately leads to the THF-dependent conversion of L-Ser to Gly and 5,10-MTHF production. However, there is no accepted catalytic mechanism for this conversion, but the available literature contains two predominant hypotheses (Figure S1)[53–56]. In a proposed two-step mechanism, after the external aldimine is formed, the β-hydroxyl of L-Ser is deprotonated by a general base, presumably Glu53 in *Tth*SHMT (Glu98 in hSHMT2, Fig. 1d), in a retro-aldol reaction to promote the release of formaldehyde. The transient formaldehyde intermediate then undergoes a nucleophilic attack by the endocyclic N5 atom of THF, likely without dissociating from the active site. A histidine residue (His122 in *Tth*SHMT, His171 in hSHMT2) has also been suggested as a possible general base in this mechanism[53,57,58]. In the competing one-step mechanism, Cβ of L-Ser within the external aldimine undergoes a direct nucleophilic attack by THF N5, facilitating α-elimination. In this reaction mechanism, protonation of Glu53 is obligatory as it acts as a general acid to protonate the hydroxyl to dehydrate Cβ[55,56,59,60]. Interestingly, within the one-step mechanism, the reaction has been proposed to proceed either by a nucleophilic displacement route through an intermediate having a covalent bond between L-Ser Cβ and THF N5 that links the external aldimine to THF, or by a concerted mechanism in which the Cβ-N5 bond formation and Cβ-Cα bond breakage occur simultaneously in the transition state to generate an intermediate product, 5-hydroxymethyl-THF. Furthermore, additional protonation/deprotonation events must occur in both proposed mechanisms that involve the THF molecule and other active site residues for the reaction to reach the final products,

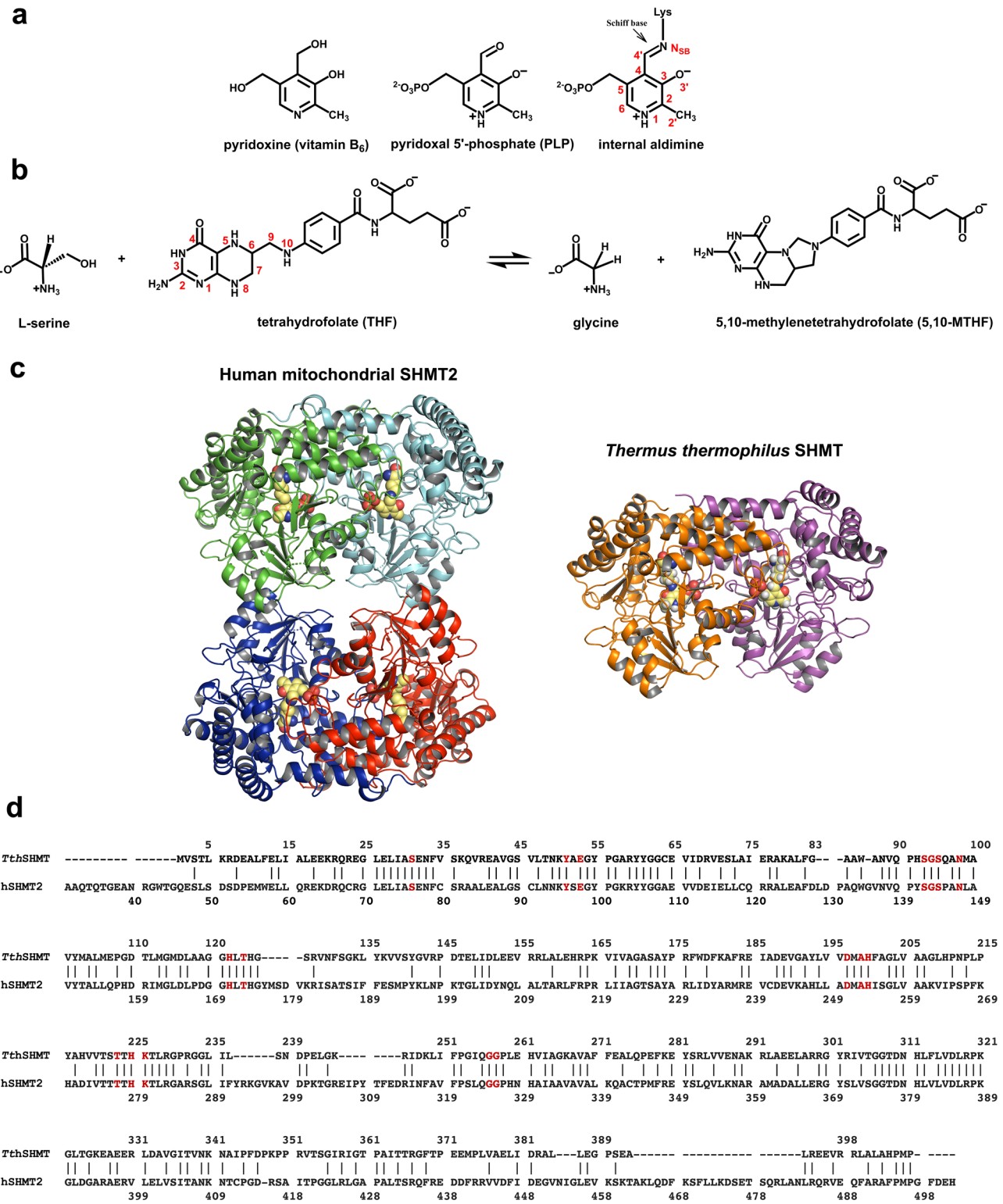

**Fig. 1 SHMT catalyzed reaction, enzyme 3D structure, and sequence alignment of *Tth*SHMT and hSHMT2. a** Chemical structures of vitamin $B_6$, PLP, and the internal aldimine. Atom labels are given on the internal aldimine structure. **b** THF-dependent conversion of L-Ser to Gly catalyzed by SHMT. **c** Overview of the overall fold of SHMT proteins from human mitochondria (hSHMT2) and *Thermus thermophilus* (*Tth*SHMT). Individual protomers are depicted in different color schemes and PLP cofactors are shown with CPK representation. hSHMT2 exists as a homotetramer made up of obligate dimers, whereas *Tth*SHMT is a homodimer. **d** *Tth*SHMT and hSHMT2 sequence alignment. Active sites are conserved between the two enzymes, highlighted in red. The sequence identity is 41%.

Gly and 5,10-MTHF. It is also known that SHMT can convert some β-hydroxyamino acids, such as L-threonine and β-phenylserine, in the THF-independent fashion to afford corresponding aldehydes[17,55,61], which would presumably require the same general base as in the THF-dependent reaction to deprotonate the hydroxyl of the β-substituted amino acid within the respective external aldimine. Taken together, the described THF-dependent SHMT catalytic mechanisms that have been proposed each require as many as six general acid-base groups, although some of them would likely be involved multiple times throughout the reaction pathway, possibly switching their roles, and thereby reducing the number to two or three[54,55]. Discriminating between these reaction mechanisms and pinpointing the general acids and bases involved will require knowledge of the protonation states of the enzyme active site residues, PLP, and THF at different reaction stages – details that can be divulged through neutron crystallography.

PLP-bound hSHMT2 forms hexagonal, rod-shaped crystals, space group P6₅22, with long cell edges (>150 Å) that diffract X-rays to resolutions of 2.3–2.5 Å using synchrotron radiation at cryogenic temperatures[18,37,62,63]. Such hSHMT2 crystals are not amenable to neutron crystallography on the currently available neutron crystallographic beamlines where neutron beam fluxes are significantly weaker than the X-ray beam fluxes at modern synchrotron facilities[64]. Like many metabolic enzymes, SHMT is evolutionarily highly conserved, especially in the active site, enabling us to utilize SHMT from *Thermus thermophilus* (*Tth*SHMT, Fig. 1c and S2) as a model of the mitochondrial human enzyme. *Tth*SHMT and hSHMT2, both fold type I PLP-dependent enzymes, have completely conserved active sites (Fig. 1d and S3), but the crystallographic unit cell volume of *Tth*SHMT is considerably smaller than that of its human counterpart. *Tth*SHMT crystals are well-diffracting, permitting us to use them for room-temperature X-ray and neutron diffraction experiments. Here, we present a 2.3 Å joint X-ray/neutron (XN) structure of the homodimeric *Tth*SHMT in the open conformation. The XN structure depicts equivalent active sites in the two protomers with the PLP cofactor covalently bound to the catalytic Lys226 in the internal aldimine state and a sulfate ion filling the substrate binding site in each protomer. For direct comparison with the joint XN structure, we also determined a 2.5 Å room-temperature X-ray structure of hSHMT2 in the internal aldimine state. To track an amino acid substrate within the SHMT active site cavity, we obtained a second joint XN structure by soaking the same *Tth*SHMT crystal with deuterated L-Ser (L-Ser-d₇). This structure revealed L-Ser-d₇ bound in a solvent-exposed peripheral substrate binding site at the entrance to the active site cavity in an apparent *pre*-Michaelis complex, with the sulfate anion continuing to block the L-Ser-d₇ entry deeper into the active site cavity. We further tracked the substrate through the active site cavity by obtaining a room-temperature X-ray structure of a *pseudo*-Michaelis complex by soaking a *Tth*SHMT crystal with D-Ser, a non-reactive enantiomer of L-Ser, which displaced the sulfate anion in protomer A occupying the cationic substrate binding site. Positions of H atoms were revealed by the nuclear density maps in the XN structures, thereby allowing us to accurately assign the protonation states and electrical charges for all amino acid residues, L-Ser substrate, and the PLP. By directly observing H atom locations and tracking the substrate positions, our study provides a unique atomic-level understanding of the SHMT active site that sheds new light on the enzyme's catalytic mechanism by eliminating speculation on the roles of many active site residues. We believe this knowledge can be also employed to inform the design of anticancer drugs targeting hSHMT2.

## Results

**Accurate map of protonation states in substrate-free *Tth*SHMT**. SHMTs from prokaryotic sources function as biological homodimers, whereas eukaryotic SHMTs exist as homotetramers composed of a dimer of dimers[17,18]. *Tth*SHMT crystallizes as a dimer with virtually equivalent active sites in protomers A and B, as seen in the cryogenic X-ray structure (PDB ID 2DKJ, unpublished). Similar to the other fold type I PLP-dependent enzymes, each protomer of *Tth*SHMT can be divided into large and small domains[1,65]. The large domain (residues 33–284) is composed of a seven-strand mixed β-sheet surrounded by α-helices in an α/β/α sandwich fold (Fig. S4). The small domain (residues 7–32 and 285–407) is made up of residues from both the N- and C-termini and the structure is dictated by anti-parallel β strands and α-helices in a two-fold α/β sandwich. The small domain is primarily responsible for establishing inter-subunit contacts[54,66,67]. Each active site is located at the corresponding dimer interface and is constructed from residues of both protomers.

To visualize the active site of SHMT at the atomic level of detail, we obtained a 2.3 Å neutron crystallographic structure of substrate-free *Tth*SHMT in the internal aldimine state jointly refined with 2.0 Å X-ray diffraction data (Table S1, Supplementary Data 1, PDB ID 8SUJ). We were unable to obtain neutron diffraction data from the P6₅22 crystals of hSHMT2 because they diffracted X-rays at a synchrotron only to 2.5 Å (Table S2). Therefore, to directly compare hSHMT2 with the joint XN structure of *Tth*SHMT, we also determined a 2.5 Å room-temperature X-ray structure of hSHMT2 in the internal aldimine state (Supplementary Data 2, PBD ID 8SSJ), because all previous hSHMT2 X-ray structures were done at cryogenic temperatures[18,37,62,63]. We accurately mapped the H atom positions (observed as D atoms), revealing the active site protonation states, H-bonding networks, water molecule orientations, and consequently electrostatics. Both diffraction datasets were collected at room temperature and pH of 5.5 from the same H/D exchanged protein crystal (Table S1). In *Tth*SHMT, PLP is covalently bound to Lys226 (Lys280 in hSHMT2) through a Schiff base linkage with the side chain ζ-amino group forming the internal aldimine. Because the active site structures are identical within the experimental coordinate error of 0.3 Å in the protomers A and B of the *Tth*SHMT homodimer, we selected protomer B for our structural analyses below, unless stated otherwise. The pyridine nitrogen, N1, of the PLP cofactor, is protonated. Thus, it is positively charged (Fig. 2a). N1 participates in an H-bond with the negatively charged side chain carboxylate of Asp197 (Asp251 in hSHMT2) conserved in all fold type I PLP-dependent enzymes at an O···D distance of 1.7 Å. H-bond distances observed in the joint XN structures reported here are given henceforth as the distance between a D (or H) atom of an H-bond donor and a heavy atom of an H-bond acceptor unless otherwise specified. The Asp197 side chain is correctly situated to interact with PLP's N1 by a network of H-bonds with Asn98 (Asn147 in hSHMT2), His125 (His174 in hSHMT2), and the backbone amide ND of Ala199 (Ala253 in hSHMT2). The pyridine N1 protonation has a significant role in PLP-dependent catalysis, increasing the efficiency of electron delocalization to the pyridine ring and leading to the stabilization of negatively charged reaction intermediates, but the enzyme microenvironment near N1 regulates its protonation state[68,69]. Protonated N1 was also observed in the joint XN structure of aspartate aminotransferase (AAT), another fold type I PLP-dependent enzyme[44,70], where it is similarly H-bonded to a conserved aspartate residue. Because the active sites of *Tth*SHMT and hSHMT2 are conserved we propose that N1 is also protonated in

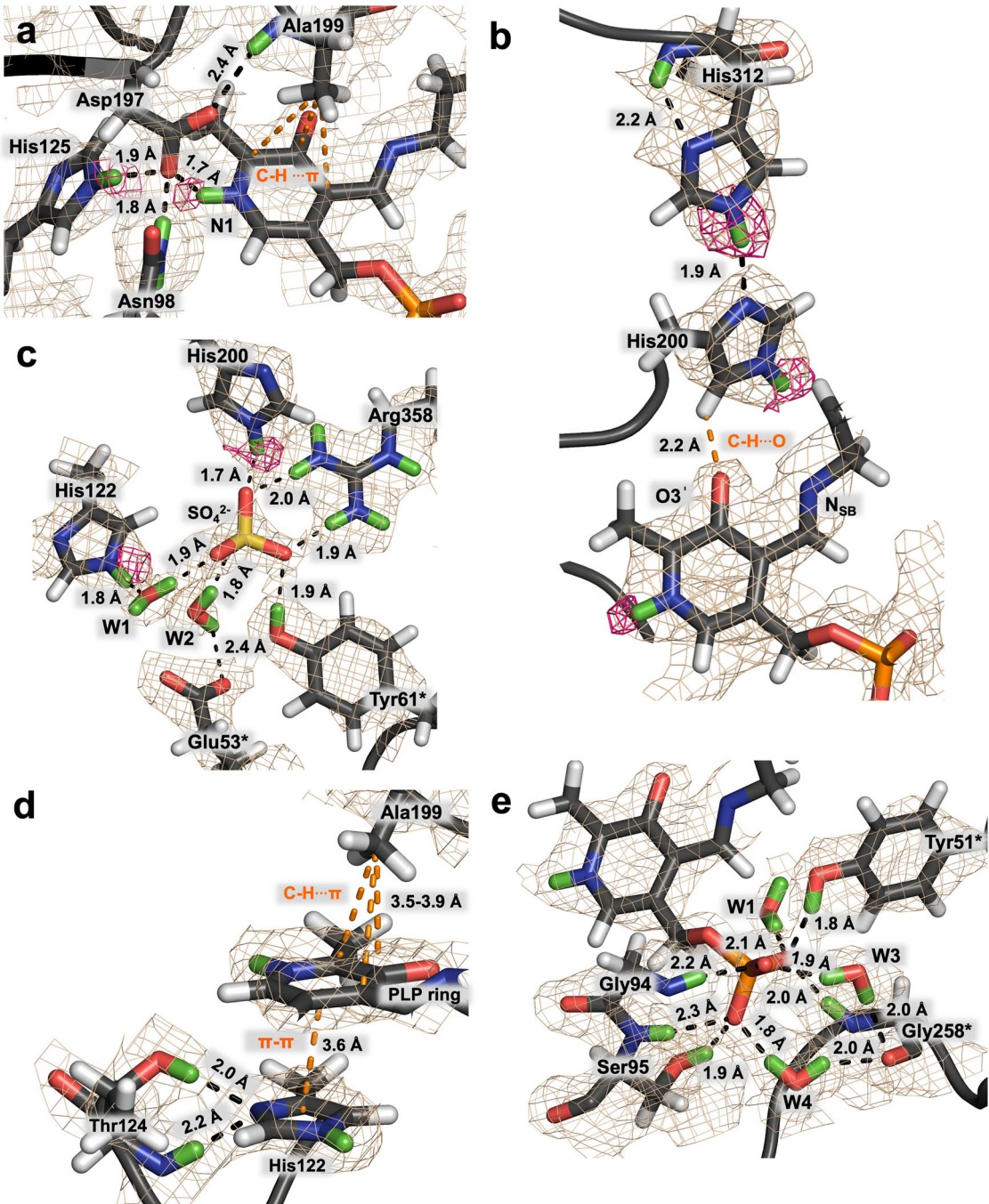

**Fig. 2 Neutron scattering length density maps for *Tth*SHMT active site residues for the joint XN internal aldimine structure.** The $2F_O\text{-}F_C$ neutron scattering length density contoured at $1\sigma$ level is depicted by wheat mesh. The omit $F_O\text{-}F_C$ difference neutron scattering length density map contoured at 2 $\sigma$ level is represented by magenta mesh. D atoms are shown in green, and H atoms are white. **a** N1 of the PLP cofactor is protonated (positively charged), but Schiff base $N_{SB}$ is not (neutral). An H-bonding network with Asn98, His125, and Ala199 fixes the position of deprotonated Asp197 (negatively charged) that forms an H-bond with N1. Above PLP, the side chain of Ala199 further stabilizes PLP position through C-H···π interactions. **b** O3' of PLP is deprotonated with a partial negative charge stabilized through an unconventional C-H···O bond with His200. A network of H-bonds made by neutral His200 and His312 connect O3' to the main chain amide ND of His312. **c** A sulfate ion accommodates the amino acid substrate binding site, and its position is stabilized through several H-bonds. **d** The pyridine ring of PLP is sandwiched between His122 and Ala199 making π-π stacking and C-H···π interactions. His122, positioned on the *re* face of the cofactor, is neutral and its imidazole group accepts D atoms from the side chain and amide backbone of Thr124 in a bifurcated H-bond. **e** The phosphate group of PLP binds in the phosphate binding pocket made up of the side chains of Tyr51* and Ser95, the amide backbone of Gly94, Ser95, and Gly258, and three crystallographic waters.

hSHMT2, participating in an H-bond with Asp251 having the N···O distance of 2.9 Å (Fig. S5).

The Schiff base nitrogen, $N_{SB}$, of the PLP cofactor is found to be not protonated, thus neutral, and the C=N bond is rotated by 35° and 29° above the plane of the pyridine ring on the *si* face of the active site in protomers A and B, respectively. $N_{SB}$ makes no interactions with the active site residues. In hSHMT2, the Schiff base rotates to a much greater extent, with the corresponding

dihedral angles in protomers A and B of 67° and 73°(Fig. S6). The Schiff base out-of-plane rotation appears to be the attribute of the $N_{SB}$ neutrality as was also observed in the joint XN structure of AAT[44]. Therefore, the above considerations point to the high probability of $N_{SB}$ atom not being protonated in hSHMT2 either. Interestingly, in the high-resolution cryogenic X-ray diffraction structure of TthSHMT (PDB ID 2DKJ) this torsion angle is reduced to 26° and 15° in protomers A and B, respectively. Protonation of $N_{SB}$ promotes the Schiff base to be more co-planar with the pyridine ring due to an H-bond formation with the phenolic oxygen O3′, as demonstrated by previous DFT calculations[9,44,71]. It is thus reasonable to suggest that $N_{SB}$ may be protonated at lower temperatures, pointing to the possible temperature dependence of the Schiff base protonation state. To gain more insight into the energy profile of the Schiff base rotation around the pyridine ring, we performed DFT calculations on the PLP-Lys model (Fig. S7). Indeed, when $N_{SB}$ is not protonated the potential energy surface is shallow with two local energy minima at ~21° and −21° in agreement with the observed dihedral angles in the joint XN structures. The global minimum occurs when C=N bond is rotated away from O3′ and is coplanar with the pyridine ring, a geometry that cannot be achieved in the protein due to geometric constraints of Lys being connected to other residues. In sharp contrast, the protonated Schiff base shows a very different potential energy profile, with the global minimum found for the C=N bond fully coplanar with the pyridine ring facing the deprotonated O3′.

The nuclear (neutron scattering length) density near the phenolic oxygen O3′ indicates the absence of a D atom. Therefore O3′ is observed as a deprotonated phenolate in the current joint XN structure. The resulting negative charge on O3′ is delocalized through resonance with the protonated pyridine ring. Both $2F_O$-$F_C$ and omit $F_O$-$F_C$ nuclear density maps clearly show that O3′ engages in an unconventional C-H···O hydrogen bond with Cδ2 of His200 imidazole (His254 in hSHMT2) with the distance of 2.0 Å (Fig. 2b). This orients the protonated Nε2 of His200 to donate a 1.7 Å H-bond with the sulfate ion ($SO_4^{2-}$) located in the substrate binding pocket near Arg358 (Arg425 in hSHMT2) (Fig. 2c). The sulfate is present in the crystallization solutions and its presence in the structure is an artifact of the crystallization conditions. In contrast to the current joint XN structure, in some previously published X-ray crystal structures of SHMT from other species, the His residues equivalent to TthSHMT His200 were incorrectly modeled to participate in H-bonds with O3′ with their Nδ1 atoms[53,54]. In TthSHMT, His200 is found to be singly protonated, thus neutral, its Nδ1 accepting a 1.9 Å H-bond with Nε2 of His312 (His380 in hSHMT2). Furthermore, His312 Nε2 faces its own main chain amide ND to form a 2.2 Å H-bond and is also singly protonated and neutral. Thus, a network of unconventional C-H···O and conventional N-H···N H-bonds (Fig. 2b) that connects the negatively charged PLP O3′ to the His312 main chain ensures the neutral charge of both His200 and His312, which would undoubtedly remain neutral at the physiological pH of 7.4. The H-bond network connecting PLP O3′ to the main chain NH of His380 is identical in hSHMT2, suggesting the same protonation states of O3′, His254, and His380 in the human enzyme as observed in TthSHMT.

Found on the re face of PLP, the sulfate ion mimics substrate binding, occupying the presumed substrate binding pocket. The sulfate position is stabilized by an H-bonding network consisting of Ser31 (Ser76 in hSHMT2), His200, Arg358, Tyr61* (Tyr106* in hSHMT2, the asterisk denotes that the residue belongs to the other protomer), and two water molecules (Fig. 2c). The sulfate participates in a salt bridge with Arg358, with H-bond distances of 1.9 Å and 2.0 Å. Ser31 and Tyr61* donate their hydroxyl D atoms in 2.0 and 1.9 Å H-bonds with the sulfate, respectively.

The two water molecules coordinate the sulfate with H-bond distances of 1.8 and 1.9 Å and link it to Glu53* (Glu98 in hSHMT2). Glu53, a conserved active site residue that is proposed to take part in the THF-dependent cleavage of L-Ser as a general acid/base catalyst[55,56,58,72], is deprotonated and positioned on the re face of PLP near the substrate binding pocket (Fig. 2c). We believe Glu98 in hSHMT2 is also deprotonated because the electrostatic environment is the same in the substrate binding pocket of the human enzyme. In our room-temperature structure of hSHMT2, the substrate binding pocket is occupied by a chloride anion, Cl⁻. The presence of Cl⁻ can be physiologically relevant as intracellular Cl⁻ concentration can be ~20–100 mM[73,74]. The affinity of this pocket for anionic ligands is probably dictated by the nearby positively charged arginine residue, Arg358 in TthSHMT or Arg425 in hSHMT2. In contrast, a glycerol molecule is found in the substrate binding pocket in the X-ray structure of hSHMT2 obtained under cryogenic conditions with glycerol added as the cryoprotectant[18]. Glycerol is normally used at concentrations of ~2 M for cryoprotection of protein crystals and can undoubtedly outcompete an anion for binding in this pocket.

The PLP pyridine ring is sandwiched between Ala199 (Ala253 in hSHMT2) on the si face and His122 (His171 in hSHMT2) on the re face (Fig. 2d). The side chain of Ala199 makes C-H···π interactions with the π-conjugated system of the pyridine ring, with the heavy atom distances in the range of 3.5–3.9 Å. This residue is conserved in fold type I PLP-dependent enzymes. Site-directed mutagenesis studies suggest the residue at this position may help modulate the $pK_a$ of the internal aldimine $N_{SB}$[75]. The His122 imidazole ring sits parallel to the pyridine ring in a π-π stacking interaction with distances of 3.5–3.7 Å between the heavy atoms. His122 is singly protonated on Nε2 (Fig. 2c, d). The neutral charge of His122 is the result of the imidazole Nδ1 accepting a bifurcated H-bond with the side chain hydroxyl and main chain amide ND of Thr124 (Thr173 in hSHMT2) with the distances of 2.0 and 2.2 Å, respectively. His122 also donates a 1.8 Å H-bond with a water molecule that mediates the imidazole interactions with the sulfate and PLP phosphate group (Fig. 2c). Again, in hSHMT2 His171 and Ala253 make very similar interactions with PLP and H-bonds with adjacent residues, allowing us to postulate that the His171 protonation state is the same as that of His122 in TthSHMT (Fig. S5).

In addition to making a covalent bond with the catalytic Lys226 and an H-bond between pyridine N1 and Asp197, the PLP position in the active site is further stabilized through an intricate network of H-bonds and water-mediated interactions made by the phosphate group with the phosphate binding pocket. In TthSHMT, the phosphate binding pocket is composed of the side chains of Ser95 (Ser144 in hSHMT2) and Tyr51* (Tyr96 in hSHMT2), the main chain amide NDs of Gly94 (Gly143 in hSHMT2), Ser95, and Gly258* (Gly326 in hSHMT2), and three water molecules. The phosphate group makes 1.8 and 1.9 Å hydrogen bonds with Ser95 and Tyr51* hydroxyls, but somewhat longer 2.0–2.3 Å H-bonds with the main chain amides (Fig. 2e). The water molecules mediate the phosphate group interactions with the main chain carbonyl of Gly258*, and with the side chain of His122 mentioned above.

**L-Ser binds as a zwitterion in the *pre*-Michaelis complex.** To obtain a complex with the bound substrate, we repurposed the TthSHMT crystal from the first neutron diffraction experiment by soaking it with the deuterated serine substrate, L-Ser-d7 at 500 mM concentration. We then collected a 2.3 Å room-temperature neutron diffraction dataset on this crystal. Joint XN refinement was performed using 2.0 Å room-temperature

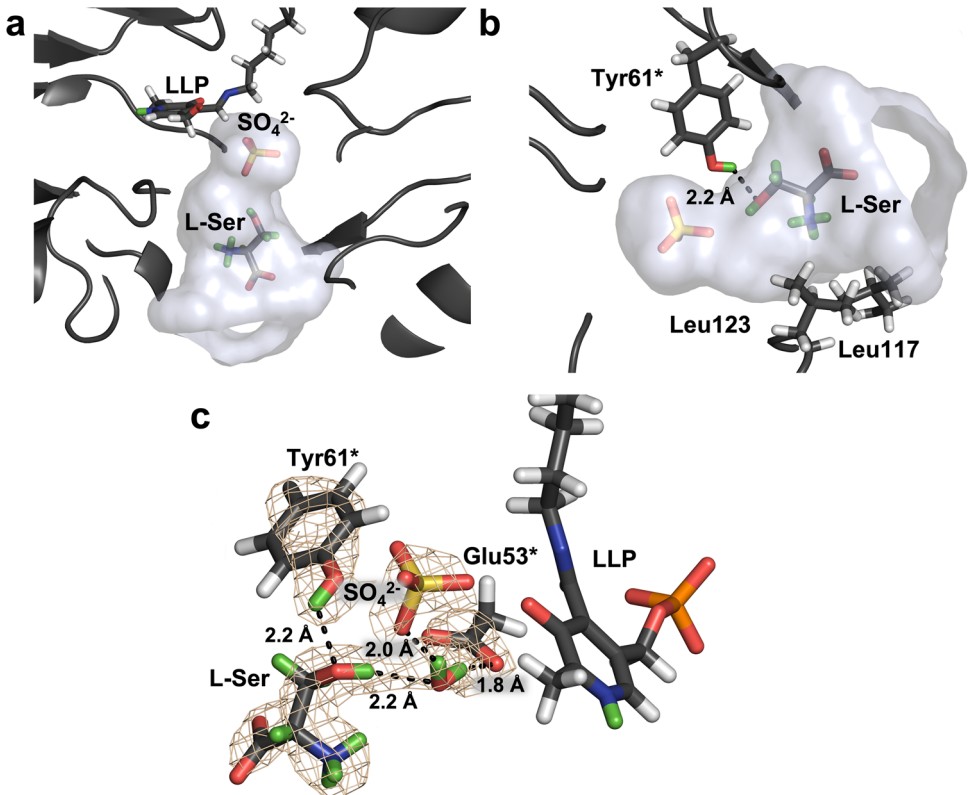

**Fig. 3 Binding of L-Ser-d₇ to the peripheral binding site within the funnel-shaped active site cavity as observed in the joint XN structure of *Tth*SHMT/L-Ser *pre*-Michaelis complex. a** L-Ser-d₇ is found at the entrance to the active site cavity in front of the sulfate ion. **b** The α-amino group of L-Ser-d₇ faces a hydrophobic patch (Leu117 and Leu123) and its carboxyl group is directed towards the bulk solvent. The side chain hydroxyl of L-Ser-d₇ makes an H-bond with Tyr61*. **c** The 2F$_O$-F$_C$ neutron scattering length density in the L-Ser-d₇ binding site contoured at 1σ level is depicted by wheat mesh. In addition to an H-bond with Tyr61*, the substrate has a water-mediated interaction with Glu53*.

X-ray diffraction data collected on the same crystal immediately after the neutron experiment. Although it was expected that L-Ser-d₇ would displace Lys226 in the Schiff base linkage to PLP to form the external aldimine, the intact internal aldimine was observed in this structure with the sulfate ions still occupying the substrate binding sites in both protomers. However, we observed L-Ser-d₇ bound in a pocket on the periphery of the active site in protomer A, with the substrate positioned in front of the sulfate ion (Fig. 3a, b; Supplementary Data 3, PBD ID 8SUI). We, therefore, call this complex, *Tth*SHMT/L-Ser, a *pre*-Michaelis complex, in which both protomers display the same open conformation observed in the substrate-free internal aldimine joint XN structure. The enzyme active site entrance resembles a funnel, with a wide opening that narrows down going deeper towards PLP. The peripheral binding site of L-Ser-d₇ is where various ligands, including THF analogs and non-THF high-affinity inhibitors such as SHIN-1, are located in the structures of SHMT from different sources[37,62,76,77]; however, amino acid substrates have not been captured previously in this site.

In protomer A of *Tth*SHMT/L-Ser, L-Ser-d₇ is observed in the zwitterionic state with the protonated α-amino group facing a hydrophobic patch made up of Leu117 (Leu166 in hSHMT2), Leu123 (Leu172 in hSHMT2), and Phe252* (Phe320* in hSHMT2), whereas the deprotonated carboxyl group is directed into the bulk solvent (Fig. 3b). The side chain hydroxyl is positioned towards the substrate binding site next to the sulfate ion. The hydroxyl group of L-Ser-d₇ donates its D atom in a weak bifurcated H-bond with a water molecule (2.20 Å), mediating its contact with Glu53*, and the sulfate ion (2.6 Å) (Fig. 3c). The hydroxyl group of Tyr61* (Tyr106* in hSHMT2) rotates away

from the sulfate ion to lose its H-bond as found in the substrate-free *Tth*SHMT structure, and now makes a 2.2 Å H-bond with the L-Ser hydroxyl oxygen. It is important to note that in the *Tth*SHMT/L-Ser joint XN structure, we found the same protonation states in the active sites of both protomers that were observed in the substrate-free structure. The N$_{SB}$ atoms of the Schiff bases are neutral, as are all the His residues, whereas the pyridine N1 is protonated. Thus, the non-covalent binding of the substrate in the *pre*-Michaelis complex does not disturb the locations of H atoms and hence the electrostatics of the active site cavity.

**D-Ser occupies the substrate binding site in a *pseudo*-Michaelis complex.** In the *Tth*SHMT joint XN structures described above, the position occupied by the sulfate ion (or Cl⁻ in hSHMT2) in the active sites is presumably the binding site for the amino acid substrate en route to generate the external aldimine intermediate. To capture a substrate bound at this site, we soaked a *Tth*SHMT crystal with D-Ser and obtained a 1.8 Å room-temperature X-ray structure of the complex (Supplementary Data 4, PBD ID 8SSY). The crystal soaking experiment produced a complex with D-Ser in protomer A replacing the sulfate, whereas protomer B retained the sulfate ion in the substrate binding site. Both protomers remained in the open conformation. D-Ser, the enantiomer of the physiological substrate L-Ser, did not proceed to react with C4′ of the internal aldimine thus generating a *pseudo*-Michaelis complex, *Tth*SHMT/D-Ser. It was not however possible to produce crystals of this complex of sufficient volume for neutron diffraction experiments.

H-bond interactions made by the D-Ser carboxylate and side chain hydroxyl groups orient the substrate enantiomer in the

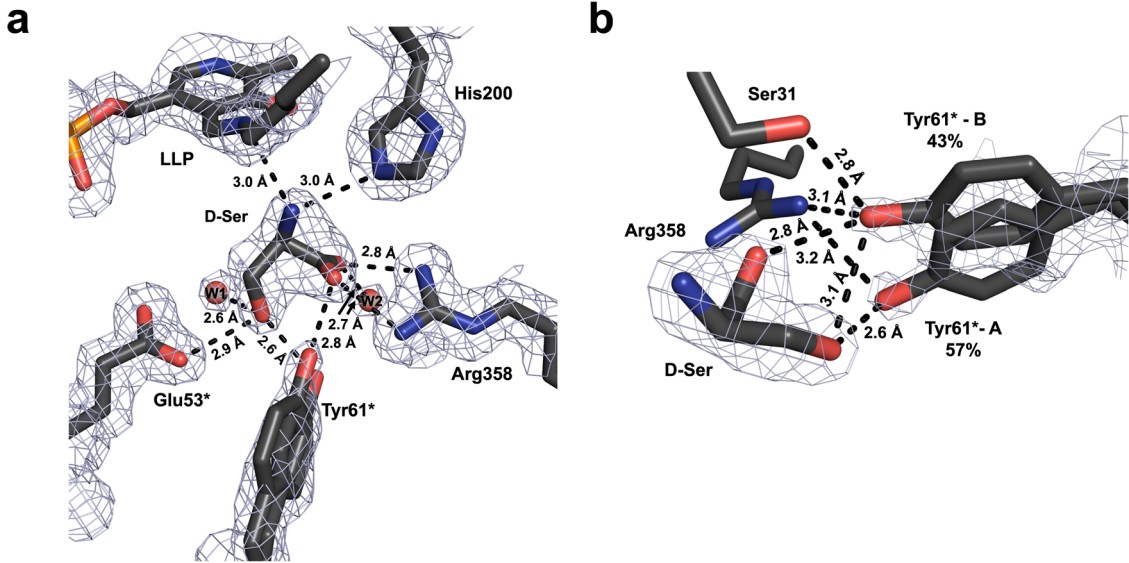

**Fig. 4 Binding of D-Ser to the substrate binding site as observed in the room-temperature X-ray structure of *Tth*SHMT/D-Ser *pseudo*-Michaelis complex.** The 2F$_O$-F$_C$ electron density contoured at 1σ level is depicted by gray mesh. **a** D-Ser makes numerous H-bonds with the active site residues. Tyr61* adopts two alternate conformations which make different substrate interactions depicted in panel (**b**). The D-Ser α-amino group is oriented away from C4′, presumably preventing the formation of the external aldimine. **b** Zoomed-in view of Tyr61* interactions with D-Ser.

active site in a specific fashion (Fig. 4a). Arg358 acts as the primary anchoring point, forming a salt bridge with the D-Ser carboxylate by making two H-bonds with the N···O distances of 2.7 Å and 2.8 Å. Tyr61* that interacts with the D-Ser hydroxyl adopts two alternate conformations (Fig. 4b). In one conformation, at 57% occupancy, the Tyr61* phenolic hydroxyl is shifted towards the hydroxyl group of D-Ser with an O···O distance of 2.6 Å. In the other conformation, at 43% occupancy, the Tyr61* phenol is rotated slightly away to form H-bonds with the D-Ser carboxylate and Ser31 side chain with both distances of 2.8 Å, resulting in a longer interaction of 3.1 Å with the D-Ser hydroxyl group. Tyr61 occupies a single conformation in each of the *Tth*SHMT joint XN structures, therefore its positional disorder in *Tth*SHMT/D-Ser is due to D-Ser binding. In the *pseudo*-Michaelis complex, Glu53* was observed within an H-bond distance of 2.9 Å to the D-Ser hydroxyl. All these H-bonds would probably be retained when L-Ser binds in the Michaelis complex before its nucleophilic α-amino group attacks C4′ of the internal aldimine for the reaction to proceed to the external aldimine. Importantly, the H-bonds with the residues corresponding to Arg358, Glu53*, and Tyr61* are also observed in the X-ray structures of the L-Ser external aldimine state for SHMTs from different species (PDB IDs: 4PFN, 6CDI)[77–79], indicating minimal structural rearrangements upon the external aldimine formation.

In the *Tth*SHMT/D-Ser complex, the D-Ser α-amino group is 3.0 Å away from N$_{SB}$ but is angled away from C4′ at a distance of 3.5 Å, possibly making the nucleophilic attack unfavorable and producing the stable *pseudo*-Michaelis complex. In contrast, D-Ser readily reacted with the internal aldimine of *Plasmodium vivax* SHMT to generate the corresponding external aldimine[78]. Because H atoms cannot be observed in the *Tth*SHMT/D-Ser X-ray structure, the protonation states of the cofactor and D-Ser cannot be determined. Therefore, we do not know whether the substrate would still be zwitterionic in the *pseudo*-Michaelis complex as observed in our joint XN structure of the *pre*-Michaelis complex. Previously, based on the XN structures of AAT in the internal and external aldimine states[44], we suggested that the protonated α-amino group of an amino acid substrate might transfer a proton to N$_{SB}$, generating the reactive nucleophilic amine, in the Michaelis complex. In *Tth*SHMT/D-Ser, the D-Ser α-amino group makes a

3.0 Å H-bond with His200 Nε2 that is protonated, indicating the amine may already be in the reactive state. Moreover, the Schiff base out-of-plane dihedral relative to the pyridine ring is flattened to 20° indicative of N$_{SB}$ being protonated and positively charged. The corresponding Schiff base dihedral in protomer B is 26°, similar to the value of 29° found in protomer B of the substrate-free *Tth*SHMT XN structure.

**Protonation of Glu residues.** Outside of the active site, we detected three protonated glutamate residues in the *Tth*SHMT homodimer (Fig. S8). The buried Glu261 is protonated in both protomers A and B on Oε2 atoms forming H-bonds of 1.7 Å and 1.6 Å, respectively, with Glu71 from the neighboring α-helix. The O···O distances of these H-bonds are 2.7 (protomer A) and 2.5 Å (protomer B). We, therefore, observed fully localized D atoms participating in short conventional H-bonds in the Glu-Glu pairs. Similar observations were made in high-resolution X-ray structures of protein DJ-1 (Lin et al. 2017) and in the neutron structure of HIV-1 protease gemdiol intermediate[52]. Conversely, in concanavalin A and HIV-1 protease/drug complexes[80,81] D atoms within Glu-Asp or Asp-Asp pairs, respectively, were found participating in the apparent low-barrier hydrogen bonds manifested by the D atoms located midway between the oxygen atoms. It is possible that the H-bonding and electrostatic environment around the carboxylic groups modulate the nature of the formed H-bond. For example, in *Tth*SHMT, positively charged Arg59 is positioned near Glu71 donating its D atoms in two H-bonds with Oε1. Therefore, Glu71 Oε1 forms a bifurcated H-bond with Arg59 and protonated Glu261, possibly driving the localization of D on Glu261 Oε2. In hSHMT2, residue 329 corresponding to *Tth*SHMT Glu261 is substituted to a smaller asparagine, which makes a water-mediated contact with Glu116 (Glu71 in *Tth*SHMT) unable to form a direct H-bond.

Another protonated glutamic acid residue, Glu65 in protomer A of *Tth*SHMT, participates in a superficial long-range, electrostatic interaction with Asp333* from the small domain of protomer B (O···O distance of ~4 Å). In protomer B, Glu65* is not protonated, curling in to make an H-bond with its own main chain amide ND, and is >5 Å away from the Asp333 carboxylate of protomer A. Asp333 is not conserved in hSHMT2, where it is

Glu401. Although glutamate has a larger side chain compared to aspartate, Glu401 is >4 Å away from Glu110* for all pairs of these residues in hSHMT2 homotetramer.

## Discussion

Many biochemical reactions proceeding inside active sites of enzymes are governed by general acid-base catalysis. The catalytic reaction is typically facilitated by proton transfer events that occur along the reaction pathway. Moreover, ligand (substrate or inhibitor) binding in the enzyme active site may be accompanied by protonation state changes that alter the electrostatic landscape of the binding site relative to its initial unbound state. As a result, to fully understand enzyme catalysis and to inform structure-based drug design it is essential to capture these protonation and deprotonation events by observing the locations and movement of H atoms before and after a ligand binds and throughout the catalytic reaction. The H atom positions determine the protonation states and electrical charges of amino acid residues and ligands, and hence electrostatics, within the enzyme active sites. In this work, we utilized the power of neutrons to directly detect and visualize the positions of H atoms in a protein structure to study the catalytic mechanism and substrate binding for TthSHMT enzyme. We extended the structural and mechanistic insights gleaned from the joint XN structures to the homologous human enzyme hSHMT2 whose active site structure and amino acid composition is conserved relative to that of TthSHMT.

The serine substrate enters the TthSHMT active site through a funnel-shaped entrance first being captured at the peripheral substrate binding site (pre-Michaelis state) and then tracked deeper to the cationic substrate binding site (Michaelis state) (Fig. 5a). We observed in the room-temperature joint XN structure of TthSHMT/L-Ser complex that L-Ser is captured in the zwitterionic state ($^+$NH$_3$-Cα-COO$^-$) at the peripheral substrate binding site that is mainly hydrophobic and exposed to the bulk solvent. The L-Ser position is stabilized by the substrate's hydroxyl making a direct H-bond with Tyr61* and a water-mediated contact with Glu53*. The Tyr61* phenolic hydroxyl rotates to donate its D in an H-bond with L-Ser from its position in the substrate-free structure, where it formed an H-bond with the sulfate anion. Previous structural and mutagenesis data suggested that Tyr61 plays a role in the interconversion of the open and closed conformational states of the active site[82]. We further tracked the substrate deeper into the active site. In the TthSHMT/D-Ser room-temperature X-ray structure, D-Ser occupies the substrate binding site that evidently has some affinity to anions, as sulfate is found in TthSHMT, and chloride is observed in hSHMT2 room-temperature structures. Therefore, this binding site, likely occupied in the Michaelis complex, can be called a cationic substrate binding site. The D-Ser α-amino group is 3.5 Å away from PLP C4′ and is expected to be positioned closer in the actual Michaelis complex with L-Ser. At the cationic substrate binding site, D-Ser is anchored by a salt bridge with Arg358 and by direct H-bonds with Glu53* and Tyr61*, the H-bonds anticipated to be conserved in the Michaelis complex to orient the L-Ser substrate correctly to carry out a nucleophilic attack on the PLP Schiff base C4′ atom. Interestingly, SHIN-1, a high-affinity inhibitor of hSHMT2, binds at the peripheral substrate binding site in the X-ray structure of E. faecium SHMT displacing the Tyr61* side chain by a bulky isopropyl group[77]. Based on our structures, we propose that the future inhibitor design could take advantage of the existence of the two substrate binding sites that provide opportunities for additional H-bonds and electrostatic interactions to be made by designed inhibitors.

In the internal aldimine state, when the SHMT active site is substrate-free or L-Ser has not yet reached the cationic substrate binding site to react with the Schiff base, the PLP N$_{SB}$ atom is not protonated whereas the pyridine N1 is protonated. Identical protonation states for PLP in the internal aldimine state have been observed in the joint XN structure of AAT[44]. In disagreement with our experimental results, however, several published theoretical calculations of the SHMT catalytic mechanism used protonated, positively charged, N$_{SB}$ in the internal aldimine state[60,83,84]. In order to proceed through the transaldimination reaction, the incoming amino acid substrate needs to be deprotonated. We believe that N$_{SB}$ receives the proton from the α-amino group of the zwitterionic amino acid substrate just before or in concert with its attack on the PLP C4′ (Fig. 5b). Although the anionic α-amino group of L-Ser ($^-$NH-Cα-COO$^-$) was previously suggested as the true substrate[85], our joint XN structures are not in favor of this scenario because the only nearby residue, His200, capable of accepting an H atom from the substrate is protonated on Nε2 that would face the α-amino group.

According to the retro-aldol catalytic mechanism of SHMT catalysis, a general base should abstract a proton from the β-hydroxyl of L-Ser when it becomes part of the external aldimine. The previously proposed candidates for this role are Glu53, His122, His125, and His200[53,66,86,87]. Analysis of our joint XN structures reveals that for the THF-dependent catalysis, the best candidate for the general base is Glu53 which is found in the deprotonated, carboxylate, form with an apparently low pK$_a$. We believe that Glu53 would also act as the general base in the THF-independent reaction converting β-hydroxyamino acids to the corresponding aldehydes and Gly. Conversely, none of the three histidine residues, although neutral, can act as a general base because they are locked in H-bonds that would preclude their protonation, making them non-titratable residues at physiological pH. Specifically, His122, π-π stacked against the PLP pyridine ring, participates in H-bonds with Thr124 and its own main chain amide NH using Nδ1, whereas Nε2-H is directed towards the substrate. His125 donates Nε2-H in an H-bond with Asp197 and Nδ1 makes an H-bond with the main chain amide NH of His122. Moreover, His125 is >9 Å away from the cationic substrate binding site, too far to participate in the reaction. His200 also cannot be considered a possible catalytic base. It is part of the H-bond network that links O3′ to the His312 main chain. His200 Nε2-H faces the substrate, as mentioned above, and its Nδ1 is in an H-bond with His312 Nε2-H. Consequently, none of these histidine residues can be protonated during the SHMT catalysis. Moreover, they remained neutral even at the acidic pH of 5.5 used to grow TthSHMT crystals. Our results thus indicate that Glu53 plays the general base role to deprotonate β-hydroxyl of L-Ser (Fig. 5b), in agreement with the recent QM/MM studies which concluded that the retro-aldol mechanism is most probable for SHMT[60].

In conclusion, neutron crystallography at near-physiological (room) temperature was used to obtain all-atom structures of TthSHMT where positions of H and D atoms were accurately located revealing protonation states of key residues and, as a result, mapping electrostatics in the enzyme active site. We observed protonated (positively charged) pyridine N1, deprotonated (negatively charged) phenolic O3′, and non-protonated (neutral) Schiff base N$_{SB}$ atoms within the PLP cofactor. Interestingly, all active site His residues were found in the mono-protonated (neutral) state, participating in H-bonding networks that would prevent their protonation at physiological pH. Based on the observed protonation states in the active site, Glu53 is proposed as the best candidate for the general base catalyst to orchestrate the retro-aldol transformation of L-Ser into Gly. The pathway of the substrate amino acid serine entering the active site cavity was tracked revealing a peripheral binding site, where the amino acid exists in the zwitterionic state, and the cationic binding site, where the substrate's α-amino group is poised for the

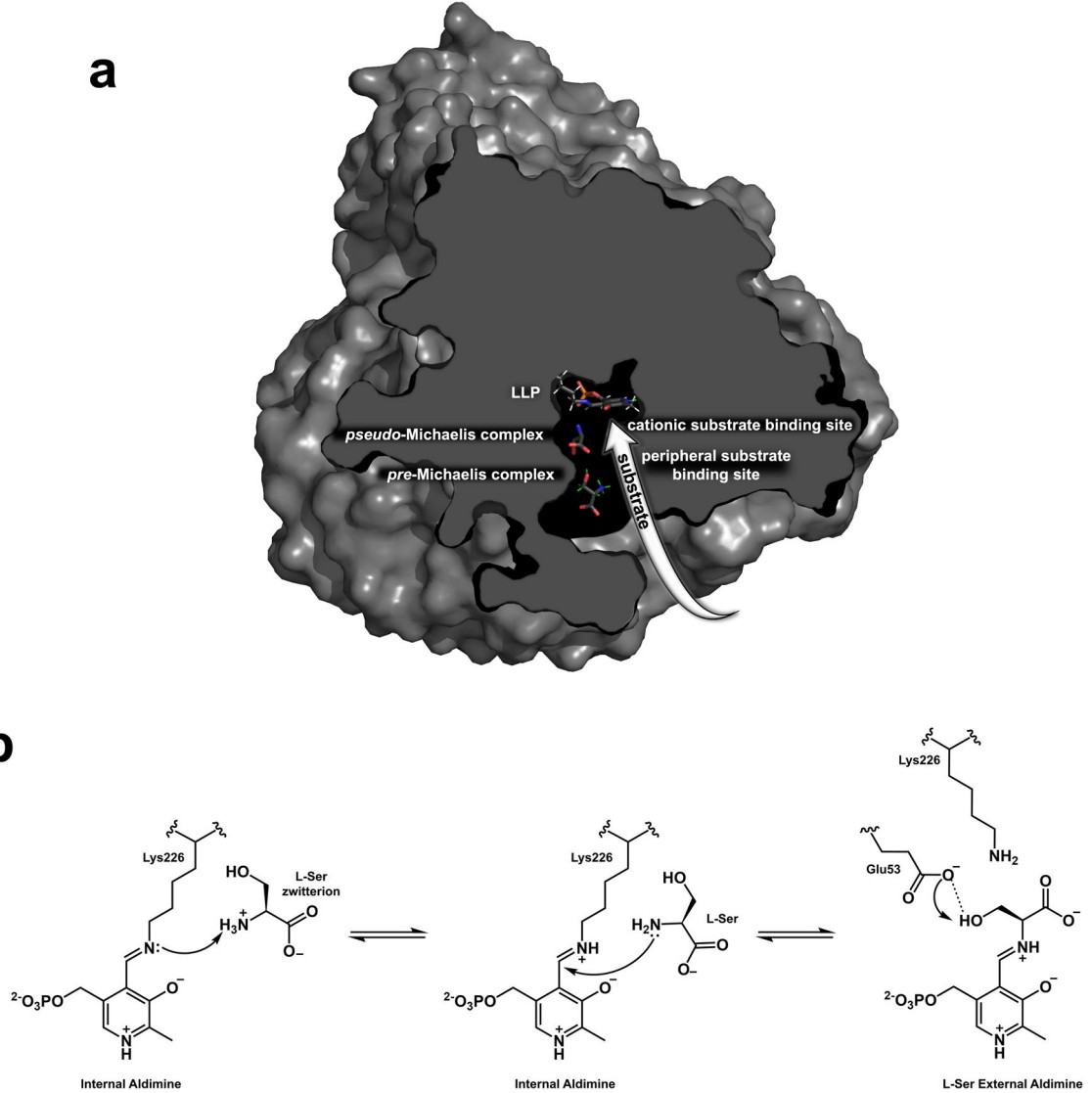

**Fig. 5 Proposed L-Ser substrate pathway through the *Tth*SHMT active site cavity and the mechanism of the external aldimine formation. a** Overview of substrate tracking into the active site of *Tth*SHMT revealed by X-ray and joint X-ray/neutron crystallography. The composite image was made by superimposing the *Tth*SHMT/L-Ser (RMS = 0.081) neutron structures and the TthSHMT/D-Ser (RMSD = 0.122) room-temperature X-ray structure with the substrate-free *Tth*SHMT. **b** Proposed mechanism for the generation of the external aldimine based on the current joint X-ray/neutron structures. Glu53 in TthSHMT is proposed to act as the general base to abstract the β-hydroxyl of L-Ser to subsequently form formaldehyde.

nucleophilic attack on Schiff base C4′ atom. Understanding the next stages in the SHMT catalysis will require mapping the H atom movements in the external aldimine with bound L-Ser and in complexes with THF or its analogs.

## Materials and methods

**General information**. Columns for protein purification were purchased from Cytiva (Piscataway, New Jersey, USA). His-tagged Tobacco Etch Virus (TEV) protease was produced in-house. Crystallization reagents and supplies were purchased from Hampton Research (Aliso Viejo, California, USA). Crystallographic supplies for crystal mounting and X-ray and neutron diffraction data collection at room temperature were purchased from MiTeGen (Ithaca, New York, USA) and Vitrocom (Mountain Lakes, New Jersey, USA). Deuterated L-Ser and hydrogenous D-Ser were purchased from Millipore Sigma (St. Louis, Montana, USA).

**TthSHMT expression and purification**. The *glyA* gene (Fig. S9) encoding SHMT enzyme from the bacterium *Thermus thermophilus* was codon optimized, synthesized, and cloned into kanamycin-resistance plasmid, pJ411 (ATUM, Newark, CA), in addition to a DNA sequence encoding for an N-terminal polyhistidine-(His₆)-tag with a 34 amino acid long linker. A TEV protease cleavage tag, ENLYFQS, was introduced at the *Tth*SHMT N-terminus sequence so that after cleavage the

enzyme sequence started from Ser3 (Fig. S9). This was done because residues Met1 and Val2 are not visible in the electron density maps in the previously deposited *Tth*SHMT structure (PDB ID 2DKJ). The plasmid was transformed into BL21(DE3) competent *E.coli* cells for expression. Transformed cells were grown in Luria-Bertani (LB) media supplemented with 50 µg/mL kanamycin antibiotic at 37 °C to an optical density of 0.8–1.0 and induced overnight with 1 mM isopropyl ß-D-1-thiogalactopyranoside (IPTG) at 22 °C (approximately 16–18 h). Induced cells were harvested by centrifugation at 5660 rpm at 4 °C, producing typical yields of 10 g of cells per 1 L of cell culture. A buffer containing 50 mM sodium phosphate pH 7.5, 500 mM NaCl, and 10 mM imidazole was used to resuspend the cell pellet, utilizing 5 mL of lysis buffer per gram of wet cell paste. The cells were stirred on ice for 30 minutes prior to mechanical sonication. The cell lysate was clarified by centrifugation at 17,000 rpm (~30,000 g) for 30 min. The supernatant was loaded onto a 5 mL HisTrap FF nickel column equilibrated with 20 mM HEPES pH 7.5, 100 mM NaCl, and 10 mM imidazole and washed with 5 column volumes (CV) of 20 mM HEPES pH 7.5, 100 mM NaCl, and 20 mM imidazole. The pure, tagged protein was eluted with 20 mM HEPES pH 7.5, 100 mM NaCl, and 500 mM imidazole in a linear gradient at relatively low imidazole concentrations (~50–60 mM). SHMT-containing fractions were pooled, and TEV protease was added to cleave the poly-histidine tag (1 mg TEV protease/100 mg of tagged protein). After room temperature overnight dialysis against 20 mM HEPES pH 7.5, 100 mM NaCl, and 1 mM EDTA, the TEV protease-treated fractions were loaded

onto the 5 mL HisTrap FF nickel column and eluted in the flow-through. Pure *Tth*SHMT, verified by SDS-PAGE, was then dialyzed overnight against 40 mM NaOAc pH 5.4 and 1 mM PLP at 4 °C, concentrated to 19 mg/mL, and stored at −30 °C in the presence of 20% (v/v) glycerol.

**hSHMT2 expression and purification.** The *SHMT2* gene encoding the SHMT enzyme from human mitochondrion (residues 37-504, Uniprot ID P34897, Fig. S10) with mitochondrial leader sequence deleted was codon optimized, synthesized, and cloned into kanamycin-resistance plasmid, pJ411 (ATUM, Newark, CA). The DNA sequence encoding for a 35 amino acid long linking sequence containing an N-terminal His$_6$-tag and the TEV protease cleavage tag so that after cleavage the enzyme sequence starts with Gly37 (Fig. S10). The plasmid was transformed into BL21(DE3) competent *E. coli* cells for expression and grown in LB media supplemented with 50 μg/mL kanamycin antibiotic at 37 °C. The culture was grown to an optical density of 0.8–1.0 and induced overnight with 1 mM IPTG at 22 °C (approximately 16–18 h). Induced cells were harvested by centrifugation at 5660 rpm at 4 °C. Using 5 mL of lysis buffer per gram of wet cell paste, the cell pellet was resuspended in a buffer containing 20 mM HEPES pH 7.5, 500 mM NaCl, and 10 mM imidazole. Lysozyme was added at 0.1 mg/mL as the cells were stirred on ice for 30 min and then subsequently mechanically sonicated. The lysates were clarified by centrifugation at 30,000 *g* for 30 min and then loaded onto a 5 mL HisTrap FF nickel column equilibrated with 20 mM HEPES pH 7.5, 500 mM NaCl, and 10 mM imidazole. The column was washed with 20 mM HEPES pH 7.5, 500 mM NaCl, and 20 mM imidazole, and tagged hSHMT2 was eluted with a linear gradient of 20 mM HEPES pH 7.5, 500 mM NaCl, and 500 mM imidazole. TEV protease was added to the pooled hSHMT2-containing fractions in order to cleave the poly-histidine tag (1 mg TEV protease/100 mg of tagged protein). The sample was dialyzed overnight at room-temperature against 20 mM HEPES pH 7.5, 250 mM NaCl, and 1 mM EDTA. The TEV protease-treated fractions were loaded onto a 5 mL HisTrap FF nickel column and eluted in the flow-through. Pure hSHMT2, verified by SDS-PAGE, was then dialyzed overnight against 20 mM HEPES pH 7.5, 300 mM NaCl, and 1 mM PLP at 4 °C and concentrated to 18 mg/mL for crystallization setups. hSHMT2 was crystallized in 50 mM Tricine pH 8.2–8.4 and 11-13% PEG 3350 in sitting drop vapor diffusion experiments.

**Crystallization and H/D-exchange.** Aliquots of pure *Tth*SHMT were thawed and dialyzed against 40 mM NaOAc pH 5.4 and 1 mM PLP to remove the glycerol. *Tth*SHMT (19 mg/mL) was crystallized in 40 mM NaOAc pH 5.5, 1 M (NH$_4$)$_2$SO$_4$, and 0.5 M Li$_2$SO$_4$ in sitting drop vapor diffusion experiments, producing showers of crystals and some crystal aggregates. Several crystal aggregates were crushed in their crystallization drops to create microcrystals for microseeding experiments. Large, single crystals for neutron diffraction were grown using a streak seeding method in 9-well glass plates and sandwich box setups. Specifically, after large crystallization drops were set up, a seeding tool from Hampton Research was dipped into the crushed crystals and then dipped quickly into the new crystal drops to transfer microcrystals. In this approach, only a few, but larger, crystals per drop of *Tth*SHMT grow. A crystal of ~2 mm³ in volume suitable for neutron diffraction was mounted in a 2 mm-inner diameter quartz capillary with a liquid plug of 40 mM NaOAc pH 5.5, 1.0 M (NH$_4$)$_2$SO$_4$, and 0.5 M Li$_2$SO$_4$ prepared in 100% D$_2$O to perform H/D-vapor exchange. The same crystal was used for both neutron crystallographic experiments, in the absence and presence of substrate L-Ser-d$_7$. To prepare the crystal for soaking with L-Ser, the quartz capillary was unsealed, and the H/D-exchange liquid plug was removed. The capillary was then filled with 500 mM deuterated L-Ser, 40 mM NaOAc pH 5.5, 1.0 M (NH$_4$)$_2$SO$_4$, and 0.5 Li$_2$SO$_4$ left to soak overnight. The following day, the soaking solution was removed and replaced with a liquid plug of 40 mM NaOAc pH 5.5, 1.0 M (NH$_4$)$_2$SO$_4$, and 0.5 Li$_2$SO$_4$ in 100% D$_2$O to perform H/D-vapor exchange. TthSHMT crystals for D-Ser soaking experiments were first transferred to a fresh drop containing 0.1 M NaOAc pH 5.5 and 15% PEG 4000, to remove excess sulfate, then moved to a drop containing 0.5 M D-Ser in 0.1 M NaOAc pH 5.5 and 15% PEG 4000. hSHMT2 at 18 mg/mL was crystallized in 50 mM Tricine pH 8.2–8.4 and 11–13% PEG 3350 in sitting drop vapor diffusion experiments producing hexagonal, rod-shaped crystals.

**Neutron diffraction data collection.** Neutron diffraction was tested at room temperature and a preliminary dataset was obtained on the LADI-DALI beamline[88] at the Institut Laue-Langevin (ILL) in Grenoble. A complete room-temperature neutron diffraction dataset for the *Tth*SHMT internal aldimine was collected on the IMAGINE[89–92] single-crystal diffractometer at the HFIR at ORNL using a neutron wavelength range of 2.8–4.5 Å. Each neutron image was composed of a 20 h exposure of the crystal held in a stationary position. The crystal was rotated along the vertical axis (Δφ = 8°) before collecting each successive image. The crystal orientation was changed three times by tilting the capillary with respect to the incident neutron beam to improve data completeness. In total, 44 neutron diffraction images were collected. Neutron diffraction data processing was performed with a version of LAUEGEN[93,94] modified to account for the geometry of the cylindrical image plate detector. The wavelength-normalization curve was determined using the intensities of symmetry-equivalent reflections at different wavelengths in LSCALE[95]. No explicit absorption corrections were applied. The data were scaled and merged in SCALA (Weiss 2001).

Room-temperature neutron diffraction data for the *Tth*SHMT-L-Ser pre-Michaelis complex were collected on the instrument MaNDi[96,97] at the Spallation Neutron Source (SNS) at ORNL. The crystal was held stationary for 20 h exposures and rotated 10° around the φ-axis before collecting the next image. All neutrons between 2 and 4.16 Å were used to collect the frames, with a total of 25 images collected. Neutron diffraction data collection on MaNDi was processed and integrated with 3D time-of-flight profile fitting in Mantid[98,99]. Wavelength normalization of the data was performed with LAUENORM[94,100] and the data were scaled and merged in SCALA[101]. Neutron data collection statistics for both datasets are shown in Table S1.

**Room-temperature X-ray diffraction data collection and structure refinement.** Room temperature X-ray diffraction data collection for *Tth*SHMT crystals was carried out on a Rigaku HighFlux HomeLab instrument equipped with a MicroMax-007 HF X-ray generator, Osmic VariMax optics, and a DECTRIS Eiger R 4 M detector at Oak Ridge National Laboratory. The data were indexed and integrated using the CrysAlis Pro software (Rigaku, The Woodlands, TX), then reduced and scaled with the AIMLESS program in the CCP4 software suite[102,103]. The *Tth*SHMT room-temperature X-ray structures were solved by molecular replacement in PHASER[104] using phases from PDB code 2DKJ. Room temperature X-ray diffraction data collection for hSHMT2 crystals was performed from a single crystal on the ID19 beamline at SBC-CAT using a Pilatus3 × 6 M detector at the Advanced Photon Source (APS). X-ray diffraction data were integrated and scaled using the HKL3000 software suite[105]. To minimize radiation damage to the hSHMT2 crystal, the X-ray beam intensity was attenuated 40 times and the data were collected with 0.2 sec/frame rate. The radiation damage to the crystal was estimated by the HKL3000 software to be less than 5%. The hSHMT2 structure was solved by molecular replacement using PHASER[104]. The cryo-temperature X-ray structure of hSHMT2 (PDB ID 4PVF)[18] was used as a starting model. All the structures were subsequently refined against the room temperature data with Phenix.refine from the PHENIX suite[106,107] and COOT[108–110]. Geometry validation was aided by Molprobity[111]. All ligand restraints were generated with eLBOW[112] using geometry optimized by quantum mechanical calculations in Gaussian 16[113] at B3LYP/6-31 g(d,p) level of theory. Final data collection and refinement statistics can be found in Table S2.

*Joint XN refinement.* Joint XN-refinement of the *Tth*SHMT internal aldimine structure and *Tth*SHMT pre-Michaelis complex were performed using the nCNS[114,115] patch of the Crystallography & NMR Systems (CNS)[115,116] software suite for macromolecular structure determination. The refinement procedure began with a single rigid body refinement followed by a series of atomic position, atomic displacement parameters, and D atom occupancy refinements. The structures were visualized in the graphics program COOT[108–110], in between rounds of refinements to inspect side chain modeling and correctly rotate side chain hydroxyl, thiol, and ammonium groups, as well as to make appropriate H-bonding networks based on both 2F$_O$−F$_C$ and F$_O$−F$_C$ nuclear scattering length density maps. All water molecules in the model are assumed to be and refined as D$_2$O as a consequence of the H/D-vapor exchange. Because hydrogenous protein was used in this experiment, the protein was modeled with H atoms at non-exchangeable positions, that is in C-H bonds. All labile, thus exchangeable, H positions in the structure were initially modeled as D until their occupancies were refined. An individual occupancy of −0.56 is associated with the presence of pure H at that position, whereas occupancy of 1.00 is indicative of pure D, because the neutron scattering length of H is −0.56 times that of D. Before depositing the neutron structures to the PDB, coordinates of each D atom were split into two records corresponding to an H and a D partially occupying the same site, both with positive partial occupancies that add up to unity. The percent D at a specific site is calculated according to the following formula: % D = (occupancy(D) + 0.56)/1.56. Neutron refinement statistics can be found in Table S1.

**Quantum chemical calculations.** The 2D relaxed potential energy profiles of the C4′-N$_{SB}$ bond rotation around the PLP pyridine ring were calculated with a simplified model of the internal aldimine (Figure S7). The model was truncated at Cβ of the lysine-portion of the internal aldimine and the phosphate group was removed, making C5′ a methyl group. The scans were completed at B3PW91/Def2-TZVPP level of theory[117,118] and reproduced the geometry observed in the experimental methods. Starting from the torsion angle observed in the *Tth*SHMT joint XN substrate-free internal aldimine structure, the full rotation of C3-C4- C4′-N$_{SB}$ was scanned. The scan was also performed on the model with a protonated N$_{SB}$. All calculations were performed with Gaussian16[113].

**Reporting summary.** Further information on research design is available in the Nature Portfolio Reporting Summary linked to this article.

# Data availability

The structures and corresponding structure factors have been deposited into the protein data bank with the PDB accession codes 8SUJ for *Tth*SHMT (Supplementary Data 1),

8SSJ for hSHMT2 (Supplementary Data 2) 8SUI for *Tth*SHMT/L-Ser (Supplementary Data 3), and 8SSY for *Tth*SHMT/D-Ser (Supplementary Data 4). Supporting information is available online.

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

## Acknowledgements

This research at ORNL's High Flux Isotope Reactor (IMAGINE beamline) and at ORNL's Spallation Neutron Source (MaNDi beamline) was sponsored by the Scientific User Facilities Division, Office of Basic Energy Sciences, U.S. Department of Energy. The Office of Biological and Environmental Research supported research at ORNL's Center for Structural Molecular Biology (CSMB), a DOE Office of Science User Facility. ORNL is managed by UT-Battelle LLC for DOE's Office of Science, the single largest supporter of basic research in the physical sciences in the United States. X-ray crystallographic data were in part collected at Argonne National Laboratory using Structural Biology Center (SBC) beamline ID19 at the Advanced Photon Source. Use of the Advanced Photon Source, an Office of Science User Facility operated for the U.S. Department of Energy (DOE) Office of Science by Argonne National Laboratory, was supported by the U.S. DOE under Contract No. DE-AC02-06CH11357. The authors thank the Institut Laue Langevin (beamline LADI-DALI) for awarded additional neutron beamtime. This research was supported by a grant from NIH-GMS (R01GM137008) to R.S.P. and A.K.

## Author contributions

R.S.P. and A.K. designed the study. V.N.D., C.C., M.H., A.C., O.G., K.L.W. and R.S.P. expressed and purified the proteins. V.N.D., C.C., O.G., and R.S.P. crystallized the proteins. V.N.D., M.H. and A.K. collected X-ray diffraction data. V.N.D. and A.K. reduced X-ray data and refined the structures. V.N.D., M.P.B., and A.K. collected and reduced neutron diffraction data. V.N.D. and A.K. refined joint XN structures. V.N.D. carried out quantum chemical calculations. V.N.D., R.S.P., and A.K. wrote the paper with help from all co-authors.

## Competing interests

The authors declare no competing interests.
