## [Peer Review File · Communications Chemistry]

Reviewers' comments:

Reviewer #1 (Remarks to the Author):

The manuscript by Drago et al. describes structural elucidation of protonation states in serine hydroxymethyltransferases from *Thermus thermophilus* and humans. Four crystal structures are presented: two joint X-ray/neutron structure of TthSHMT in the internal aldimine state, one without and one with substrate L-Ser. A third X-ray only structure of the *Thermus thermophilus* enzyme in complex with D-Ser is described. Finally, a further X-ray only structure of the human SHMT2 at room temperature is presented for comparison. Quantum chemical calculations have been done to confirm the torsion angles observed for the Schiff base bond as a function of protonation state. The main conclusion of the work is that, of the two reaction mechanisms proposed to date, the retro-aldol mechanism involving abstraction of a proton from the side chain of serine by Glu53 is the more probable, given the protonation states of residues in the active site. The neutron data also help revise the orientation of critical histidine residues and resolve the protonation states of the cofactor.

The results are solid and worthy of publication. I have a few questions relating to the experimental details and the presentation.

Presentation

There is a detailed description of the two proposed reaction mechanisms in the introduction that would benefit from being shown as reaction schemes (possibly as supplementary information) for non-experts in this particular enzyme. It would also help the reader follow the discussion starting on page 17.

Not enough mention is given to the existing crystal structure of TthSHMT in the internal aldimine state determined by Kai et al. in a Japanese structural genomics initiative, with PDB ID 2DKJ. Many of the overall descriptive features of the structure and the interaction of PLP with the binding pocket are presumably also seen in that structure. No paper was published around this structure but it should be acknowledged earlier in the text than it is. For example, mention it on page 7 on the second-last line where the “virtually equivalent active sites in protomers A and B” are described.

I think the authors have missed an opportunity to model the interactions of L-Ser in the true active site. They write that the amino group is “presumably closer” to the PLP moiety in L-Ser. If the authors believe in the interactions of the carboxyl group and the side chain, it should be a simple matter of exchanging the H atom on the C-alpha for an amino group.

It's not clear for a non-expert on SHMTs whether such a pre-Michaelis binding site has been seen in any other structure. The authors should make this clearer.

Experimental details

D-Ser is clearly able to displace SO₄ from one of the active sites because the crystals were transferred to a stabilising buffer with no SO₄. My guess is that this was not possible for the crystals used for neutron diffraction due to their size, but it would be good if this were made clearer.

Could it be that the difference in torsion angle of the Schiff base seen in hSHMT2 relative to TthSHMT is due to some kind of photoreduction event? The hSHMT2 data were collected at a synchrotron? The authors should describe what measures (if any) were taken to minimise radiation damage. Did they calculate the total dose to the crystal in Gray?

The crystallisation experiments are not sufficiently well described. Given the difficulty of making large crystals for neutron diffraction, it's not enough to write "... were grown using a streak-seeding method." The process should be described step for step. The crystal size should also be described. If I did my calculations correctly, the data collection took 36 days at IMAGINE and 20 days at MaNDi, so it's important for comparison of different neutron instruments to know how large the crystal was.

Various small things

I think "H bond..." should be written "H-bond..." throughout the manuscript.

I advise to avoid writing "phosphates" or "waters" and use rather "phosphate groups" and "water molecules"

Definite articles are missing in several places, e.g. page 9 7th and 8th last lines, in the title on page 12, page 13 line 6

Page 3 line 13: there is an extra colon inside the parentheses

Page 5 line 15-16: I would say that assigning protonation states is "often critical", but not always, for inhibitor design. Also, the cited paper by Manzoni et al. is about a carbohydrate recognition domain, not an enzyme, so I would write "understanding the mechanism of catalysis or molecular recognition".

Page 6 line 2: add a comma after "THF"

Page 6 2nd paragraph line 3: Why is a 22-year old paper relevant for neutron crystallography on currently available beamlines?

Page 8, 5th last line: add a comma after "catalysis"

Page 9, line 11: "... not being protonated either."

Page 9, 5th last line: "for the C=N bond fully coplanar with the pyridine ring" would be better

Page 10, paragraph 2 line 2: Refer to Figure 1c at the end of the first sentence, and again on the second-last line.

Page 11 line 2: Why not show the chloride ion in a supplementary figure?

Page 11 2nd paragraph line 11: "donates ... to". Is the orientation of the water molecule well defined? It's not totally clear from the figure.

Page 12: Mention the relative concentrations of L-Ser and sulphate. One can find them in the experimental section but they are relevant here I think.

Page 13, last line: give the PDB IDs for the structures referred to as well as the citations.

Page 22: I personally don't think it's necessary to cite all old papers for a given software suite when new ones that give a general description of the same suite are published, but this is a matter of taste.

Likewise, consider how many papers about IMAGINE it is relevant to cite.

Reviewer #2 (Remarks to the Author):

Drago and coworkers present a well-written manuscript detailing the PLP-dependent enzymatic reaction

of Serine hydroxymethyltransferase (SHMT), which is an important enzyme for one-carbon metabolisms that is an essential pathway for constructing fundamental biomolecules necessary for prokaryotic and eukaryotic cell growth and proliferation. Additionally, SHMT was reported as a potential target of anti-cancer, anti-malaria, and anti-bacterial agents. In this study, authors employ X-ray/neutron structures and a room-temperature X-ray structure to understand the PLP-dependent enzymatic reaction of SHMT. Based on these data, authors elucidate the protonation states of TthSHMT and suggest the binding process of Ser. Additionally, analysis of their joint XN structures reveals the best candidate for the general base. This research is important not only to understand the details of various PLP-dependent enzymatic reactions but also to develop novel SHMT inhibitors. The manuscript fits well within the scope of the journal. The reviewer recommends the manuscript for publication albeit with some issues that need to be addressed.

Major Comments

1. It has been reported that SHMT catalyzes a THF-independent retro-aldol reaction when SHMT reacts with an aromatic β -hydroxyamino acid, such as β -phenylserine, to afford a corresponding aldehyde. [Sando et al. Nat. Commun., 2019, doi: 10.1038/s41467-019-08833-7]. Authors should refer to this point and analyze this reaction using your crystal data. These analyses may help us to further understand a myriad of PLP-dependent catalysis and to develop novel SHMT inhibitors, which inhibit a different reaction step from previously reported SHMT inhibitors such as SHIN-1.

2. I'm curious whether the THF-independent retro-aldol reaction reproduces in SHMT crystals.

3. In the presence of Gly and 5-methyltetrahydrofolate (Me-THF), SHMT has an absorbance at 500 nm, which derives from quinonoid formation in the ternary complex of SHMT, Gly, and Me-THF. The enzymatic reaction of SHMT proceeds via a quinonoid intermediate [P. Stover et al. J. Biol. Chem., 1991, doi: 10.1016/S0021-9258(18)52328-0]. I'm also curious to determine the protonation state of SHMT/Gly/Me-THF ternary complex. Because authors mentioned the importance of identifying the protonation states in SHMT/Ser/THF to understand the next step in the SHMT catalysis in the last sentence of the Discussion section in the manuscript. Protonation states of SHMT/Gly/Me-THF ternary complex could contribute to understanding the next steps in the SHMT catalysis.

4. PLP and L-Ser-d₇ did not react when L-Ser-d₇ was added to the D-labeled SHMT. In crystals, structural change and fluctuation of proteins are suppressed. Due to these environmental factors, L-Ser-d₇ might be unable to push sulfate ion away and react to PLP. On the other hand, I'm concern about enzymatic activity of D-labeled SHMT. Does D-labeled SHMT remain active?

5. In the Results section, only the results of experiments and their analysis should be described. Texts describing the author's considerations should be moved to the Discussion section whenever possible.

Minor comments

Page 9, Line 16; not unreasonable → reasonable

Reviewer #3 (Remarks to the Author):

In their manuscript Drago et al. present joint X-ray/neutron structures of TthSHMT and TthSHMT/L-ser complexes as well as X-ray structures of TthSHMT/D-ser and hSHMT2. The experimental mapping of protonation states within the active site presented in this manuscript has important implications for the SHMT2 catalytic mechanism of retro-aldol cleavage of L-serine into glycine.

The structural work has been competently done and the manuscript is very well written. I have some comments on the presentation of the results, and support the publication of this manuscript once the following points have been addressed:

Major points:

Page 6, line 172. I think for the reader it is useful to define here what is meant by 'scatter X-rays poorly'. i.e. more explicitly state the resolution requirements for the collection of useful neutron data. Further, 'forms hexagonal crystals' implies that the crystal morphology is hexagonal which is not always true for hSHMT2 crystallized with antifolates and/or another inhibitors. So it would be useful to state that by hexagonal you are referring to the specific space-group P6522. I would suggest that in addition to the current reference Scaletti et al. also referencing the structures of Giardina et al., (pdb: 4pvf), Ducker et al., (pdb: 5v7i) and Ota et al (pdb: 6m5o) and stating that these structures all belong to space group P6522, that all the cell edges are $>150 \text{ \AA}$, and the resolution of these structures range from 2.3 to 2.6 \AA which is not suitable for neutron crystallography.

I would also like to point out that in the PDB there is a high resolution structure of hSHMT2 (pdb id: 8aql) in the pdb, solved to 1.23 \AA resolution in the spacegroup P21 which has much shorter unit cell dimensions (58/125/134) and a 2.04 \AA structure (pdb id: 6dk3) that was solved in the spacegroup C2 which has the unit cell dimensions 95/75/75. In addition, the crystallization conditions relating to those structures do not contain sulfate, which may be useful for future experiments due to the presence of a sulfate ion in the active site of some of your structures. So it would be useful to state that the issues with collecting neutron diffraction data apply here because your particular structure belongs to the P6522 space group and that the resolution of your structure is 2.5 \AA . This particular point should be mentioned somewhere in the results/discussion.

Minor points:

Page 5, line 141. Sentence should read 'with the L-serine substrate'

Page 9, line 241. Write full name (aspartate) instead of Asp.

Page 9, line 250. Please write full enzyme name the first time the acronym AAT is used.

Page 10, line 285. I would suggest a more speculative tone here as is used in the in the abstract. i.e.

'suggesting the same protonation states'...

Page 11, line 305. 'cryo-structure' may make some readers think you are referring to a Cryo-EM structure. This needs to be rephrased so it is clear you mean crystals frozen in liquid nitrogen which had cryo-protectant added to them. I would also be good to point out somewhere in the manuscript that existing structures of hSHMT2 prior to this study all involved cryo-protected crystals, which is why your room temperature structure is particularly useful.

Page 11, line 320. 'allowing us to conclude' should be rephrased to something more speculative (see above).

Page 11, line 327. What is the equivalent residue for Ser95?

Page 12, line 346. How conserved is the peripheral binding site (i.e. the folate binding site) between TthSHMT and hSHMT2?

Page 14, line 396. Have you tried collecting joint neutron/X-ray data for TthSHMT/D-Ser?

Page 14, lines 409 and 410. Write full name (glutamate) instead of Glu.

Other points:

*It would be useful to show a superposition of the overall TthSHMT dimer with hSHMT2.

*Please include a figure with your proposed mechanism for retro-aldol cleavage of L-serine into glycine.

*Figure 5, it would be useful to depict where inhibitors/antifolates that are present in existing hSHMT2 structures bind relative to LLP and the ligands present in your structures.

Reviewers' comments:

Reviewer #1

The manuscript by Drago et al. describes structural elucidation of protonation states in serine hydroxymethyltransferases from *Thermus thermophilus* and humans. Four crystal structures are presented: two joint X-ray/neutron structure of TthSHMT in the internal aldimine state, one without and one with substrate L-Ser. A third X-ray only structure of the *Thermus thermophilus* enzyme in complex with D-Ser is described. Finally, a further X-ray only structure of the human SHMT2 at room temperature is presented for comparison. Quantum chemical calculations have been done to confirm the torsion angles observed for the Schiff base bond as a function of protonation state. The main conclusion of the work is that, of the two reaction mechanisms proposed to date, the retro-aldol mechanism involving abstraction of a proton from the side chain of serine by Glu53 is the more probable, given the protonation states of residues in the active site. The neutron data also help revise the orientation of critical histidine residues and resolve the protonation states of the cofactor.

The results are solid and worthy of publication. I have a few questions relating to the experimental details and the presentation.

Response: We appreciate the reviewer's careful consideration of our manuscript and enthusiasm towards the presented results. We agree that the results presented in our manuscript are of biochemical importance and may help advance the rational drug design efforts targeting SHMT.

Presentation

1. There is a detailed description of the two proposed reaction mechanisms in the introduction that would benefit from being shown as reaction schemes (possibly as supplementary information) for non-experts in this particular enzyme. It would also help the reader follow the discussion starting on page 17.

Response: We have made an additional supporting figure depicting the proposed catalytic mechanisms of SHMT, which is now Figure S1.

2. Not enough mention is given to the existing crystal structure of TthSHMT in the internal aldimine state determined by Kai et al. in a Japanese structural genomics initiative, with PDB ID 2DKJ. Many of the overall descriptive features of the structure and the interaction of PLP with the binding pocket are presumably also seen in that structure. No paper was published around this structure but it should be acknowledged earlier in the text than it is. For example, mention it on page 7 on the second-last line where the "virtually equivalent active sites in protomers A and B" are described.

Response: We have added the reference to the earlier (unpublished) *TthSHMT* structure where suggested.

3. I think the authors have missed an opportunity to model the interactions of L-Ser in the true active site. They write that the amino group is “presumably closer” to the PLP moiety in L-Ser. If the authors believe in the interactions of the carboxyl group and the side chain, it should be a simple matter of exchanging the H atom on the C-alpha for an amino group.

Response: We carefully considered the reviewer’s suggestion. As it turns out, modeling L-Ser at the substrate binding site is not as simple as switching H atom on D-Ser Calpha for an amino group. This procedure places the amino group in the wrong place, requiring ~180 degree rotation around the Calpha-C(carboxyl) bond to orient the amino group towards Schiff base. In addition, some positional rearrangement of L-Ser would be needed, albeit its carboxylate would still be making a salt bridge with Arg358. Thus, correct placement of L-Ser in the substrate binding site would require extensive modeling, which is beyond the scope of the current manuscript.

4. It’s not clear for a non-expert on SHMTs whether such a pre-Michaelis binding site has been seen in any other structure. The authors should make this clearer.

Response: This is the first time such a complex has been observed. We therefore added the following phrase at the end of paragraph 1, page 12: “however, amino acid substrates have not been captured previously in this site.”

Experimental details

1. D-Ser is clearly able to displace SO₄ from one of the active sites because the crystals were transferred to a stabilising buffer with no SO₄. My guess is that this was not possible for the crystals used for neutron diffraction due to their size, but it would be good if this were made clearer.

Response: Yes, this is correct. We added a clarifying sentence on page 13: “It was not however possible to produce crystals of this complex of sufficient volume for neutron diffraction experiments.”

2. Could it be that the difference in torsion angle of the Schiff base seen in hSHMT2 relative to TthSHMT is due to some kind of photoreduction event? The hSHMT2 data were collected at a synchrotron? The authors should describe what measures (if any) were taken to minimise radiation damage. Did they calculate the total dose to the crystal in Gray?

Response: We do not believe that the Schiff base torsion angle seen in the hSHMT2 structure is due to photoreduction. We added the following sentences to the Methods, page 22: “To minimize radiation damage to the hSHMT2 crystal, the X-ray beam intensity was attenuated 40 times and the data were collected with 0.2sec/frame rate. The radiation damage to the crystal was estimated by the HKL3000 software to be less than 5%.”

3. The crystallisation experiments are not sufficiently well described. Given the difficulty of making large crystals for neutron diffraction, it's not enough to write "... were grown using a streak-seeding method." The process should be described step for step. The crystal size should also be described. If I did my calculations correctly, the data collection took 36 days at IMAGINE and 20 days at MaNDi, so it's important for comparison of different neutron instruments to know how large the crystal was.

Response: We have added details of the crystallization experiments and the crystal size in Methods at the bottom of page 20, as follows: "Several crystal aggregates were crushed in their crystallization drops to create microcrystals for microseeding experiments. Large, single crystals for neutron diffraction were grown using a streak seeding method in 9-well glass plates and sandwich box setups. Specifically, after large crystallization drops were set up, a seeding tool from Hampton research was dipped into the crushed crystals and then dipped quickly into the new crystal drops to transfer microcrystals. In this approach, only a few, but larger, crystals per drop of *TthSHMT* could grow. A crystal of $\sim 2 \text{ mm}^3$ in volume suitable for neutron diffraction ...".

Various small things

1. I think "H bond..." should be written "H-bond..." throughout the manuscript.

Response. The text has been corrected to read "H-bond..." throughout.

2. I advise to avoid writing "phosphates" or "waters" and use rather "phosphate groups" and "water molecules"

Response. The text has been corrected to read "phosphate group" or "water molecule(s)" thought, except where the phrase "water-mediated" is used.

3. Definite articles are missing in several places, e.g. page 9 7th and 8th last lines, in the title on page 12, page 13 line 6

Response. The definite articles have been added.

4. Page 3 line 13: there is an extra colon inside the parentheses

Response. The extra colon has been removed.

5. Page 5 line 15-16: I would say that assigning protonation states is "often critical", but not always, for inhibitor design. Also, the cited paper by Manzoni et al. is about a carbohydrate recognition domain, not an enzyme, so I would write "understanding the mechanism of catalysis or molecular recognition".

Response. We have revised the sentence as suggested. Additionally, we note that the other citations listed at the end of the sentence refer to studies where determination of protonation states was important for inhibitor design.

6. Page 6 line 2: add a comma after “THF”

Response. A comma was added after “THF” in the designated line.

7. Page 6 2nd paragraph line 3: Why is a 22-year old paper relevant for neutron crystallography on currently available beamlines?

Response. The old paper has been removed from the citation list.

8. Page 8, 5th last line: add a comma after “catalysis”

Response. A comma was added after “catalysis” in the designated line.

9. Page 9, line 11: “... not being protonated either.”

Response. The text was edited to read, “Therefore, the above considerations point to the high probability of N_{SB} atom not being protonated in hSHMT2 either”.

10. Page 9, 5th last line: “for the C=N bond fully coplanar with the pyridine ring” would be better

Response. The text was edited to read, “with the global minimum found for the C=N bond fully coplanar with the pyridine ring facing the deprotonated O3”.

11. Page 10, paragraph 2 line 2: Refer to Figure 1c at the end of the first sentence, and again on the second-last line.

Response. We believe the reviewer meant Figure 2c. Reference to Figure 2c has been added.

12. Page 11 line 2: Why not show the chloride ion in a supplementary figure?

Response. We have added an additional panel in Figure S5 to depict the position of the chloride in the active site.

13. Page 11 2nd paragraph line 11: “donates ... to”. Is the orientation of the water molecule well defined? It’s not totally clear from the figure.

Response. Yes, the water orientation is well defined in the nuclear density map.

14. Page 12: Mention the relative concentrations of L-Ser and sulphate. One can find them in the experimental section but they are relevant here I think.

Response. Concentration of the deuterated L-Ser in the soaking solution was added.

15. Page 13, last line: give the PDB IDs for the structures referred to as well as the citations.

Response. PDB IDs and the corresponding citations have been added.

16. Page 22: I personally don't think it's necessary to cite all old papers for a given software suite when new ones that give a general description of the same suite are published, but this is a matter of taste. Likewise, consider how many papers about IMAGINE it is relevant to cite.

Response. In the Methods we aim to cite original papers for the software that is used to process diffraction data, and relevant instrument papers.

Reviewer #2:

Drago and coworkers present a well-written manuscript detailing the PLP-dependent enzymatic reaction of Serine hydroxymethyltransferase (SHMT), which is an important enzyme for one-carbon metabolisms that is an essential pathway for constructing fundamental biomolecules necessary for prokaryotic and eukaryotic cell growth and proliferation. Additionally, SHMT was reported as a potential target of anti-cancer, anti-malaria, and anti-bacterial agents. In this study, authors employ X-ray/neutron structures and a room-temperature X-ray structure to understand the PLP-dependent enzymatic reaction of SHMT. Based on these data, authors elucidate the protonation states of TthSHMT and suggest the binding process of Ser. Additionally, analysis of their joint XN structures reveals the best candidate for the general base. This research is important not only to understand the details of various PLP-dependent enzymatic reactions but also to develop novel SHMT inhibitors. The manuscript fits well within the scope of the journal. The reviewer recommends the manuscript for publication albeit with some issues that need to be addressed.

Response. We greatly appreciate the reviewer's comments, their enthusiasm towards our study.

Major Comments

1. It has been reported that SHMT catalyzes a THF-independent retro-aldol reaction when SHMT reacts with an aromatic β -hydroxyamino acid, such as β -phenylserine, to afford a corresponding aldehyde. [Sando et al. Nat. Commun., 2019, doi: 10.1038/s41467-019-08833-7]. Authors should refer to this point and analyze this reaction using your crystal data. These analyses may help us to further understand a myriad of PLP-dependent catalysis and to develop novel SHMT inhibitors, which inhibit a different reaction step from previously reported SHMT inhibitors such as SHIN-1.

Response. We now refer to the THF-independent reaction catalyzed by SHMT in the Introduction (page 6) and Discussion (page 17).

2. I'm curious whether the THF-independent retro-aldol reaction reproduces in SHMT crystals.

Response. The reviewer asks a valid question. As with the THF-dependent reaction, following the THF-independent reaction will require trapping intermediates by soaking SHMT crystals with various β -substituted amino acids as substrates. This work is being planned by the authors as a future study.

3. In the presence of Gly and 5-methyltetrahydrofolate (Me-THF), SHMT has an absorbance at 500 nm, which derives from quinonoid formation in the ternary complex of SHMT, Gly, and Me-THF. The enzymatic reaction of SHMT proceeds via a quinonoid intermediate [P. Stover et al. J. Biol. Chem., 1991, doi: 10.1016/S0021-9258(18)52328-0]. I'm also curious to determine the protonation state of SHMT/Gly/Me-THF ternary complex. Because authors mentioned the importance of identifying the protonation states in SHMT/Ser/THF to understand the next step in the SHMT catalysis in the last sentence of the Discussion section in the manuscript. Protonation states of SHMT/Gly/Me-THF ternary complex could contribute to understanding the next steps in the SHMT catalysis.

Response. Again, obtaining the ternary complex the reviewer refers to and its neutron structure will be a very interesting and important experiment and is part of our future efforts. We would like to point out here that, according to the literature, the amount of the quinonoid produced is in several percent, whereas the bulk of the crystal is believed to be in the PLP-Gly external aldimine state. Indeed, a neutron diffraction experiment should answer the question of the protonation states in SHMT/Gly/Me-THF complex.

4. PLP and L-Ser-d7 did not react when L-Ser-d7 was added to the D-labeled SHMT. In crystals, structural change and fluctuation of proteins are suppressed. Due to these environmental factors, L-Ser-d7 might be unable to push sulfate ion away and react to PLP. On the other hand, I'm concern about enzymatic activity of D-labeled SHMT. Does D-labeled SHMT remain active?

Response. *Th*SHMT was expressed in hydrogenous media, thus no deuterium atoms were incorporated into the protein at this stage. The crystals were also grown in H₂O solutions, but then H/D exchanged with solutions made in D₂O. Therefore, only exchangeable H atoms, those in OH, NH and SH groups, were replaced with D atoms. We have previously shown that partially deuterated aspartate aminotransferase maintains catalytic activity (ref. 44 in the revised manuscript). Consequently, there is no reason for the SHMT enzyme to lose activity after H/D exchange.

5. In the Results section, only the results of experiments and their analysis should be described. Texts describing the author's considerations should be moved to the Discussion section whenever possible.

Response. We carefully re-read the Results section and could not identify authors' considerations that can be moved to Discussion. The analyses and comparisons we included in the Results organically follow experimental observations and, we believe, need to remain in this section.

Minor comments

1. Page 9, Line 16; not unreasonable → reasonable

Response. The text was changed to read, "It is thus reasonable to suggest...".

Reviewer #3:

In their manuscript Drago et al. present joint X-ray/neutron structures of TthSHMT and TthSHMT/L-ser complexes as well as X-ray structures of TthSHMT/D-ser and hSHMT2. The experimental mapping of protonation states within the active site presented in this manuscript has important implications for the SHMT2 catalytic mechanism of retro-aldol cleavage of L-serine into glycine. The structural work has been competently done and the manuscript is very well written. I have some comments on the presentation of the results, and support the publication of this manuscript once the following points have been addressed:

Response. We would like to thank the reviewer for their interest in our study and for taking effort to carefully consider our manuscript.

Major points:

2. Page 6, line 172. I think for the reader it is useful to define here what is meant by 'scatter X-rays poorly'. i.e. more explicitly state the resolution requirements for the collection of useful neutron data. Further, 'forms hexagonal crystals' implies that the crystal morphology is hexagonal which is not always true for hSHMT2 crystallized with antifolates and/or another inhibitors. So it would be useful to state that by hexagonal you are referring to the specific space-group P6522. I would suggest that in addition to the current reference Scaletti et al. also referencing the structures of Giardina et al., (pdb: 4pvf), Ducker et al., (pdb: 5v7i) and Ota et al (pdb: 6m5o) and stating that these structures all belong to space group P6522, that all the cell edges are $>150 \text{ \AA}$, and the resolution of these structures range from 2.3 to 2.6 \AA which is not suitable for neutron crystallography.

Response. We have revised these sentences on page 6/7 and incorporated suggested citations, as follows: "PLP-bound hSHMT2 forms hexagonal rod-shaped crystals, space group P6₅22, with

long cell edges (>150 Å) that diffract X-rays to resolutions of 2.3-2.5 Å using synchrotron radiation at cryogenic temperatures (Giardina et al. 2015; Ducker et al. 2017; Scaletti et al. 2019; Ota et al. 2021). Such hSHMT2 crystals are not amenable to neutron crystallography on the currently available neutron crystallographic beamlines where neutron fluxes are significantly weaker than the X-ray fluxes at modern synchrotron facilities”.

3. I would also like to point out that in the PDB there is a high resolution structure of hSHMT2 (pdb id: 8aql) in the pdb, solved to 1.23 Å resolution in the spacegroup P21 which has much shorter unit cell dimensions (58/125/134) and a 2.04 Å structure (pdb id: 6dk3) that was solved in the spacegroup C2 which has the unit cell dimensions 95/75/75. In addition, the crystallization conditions relating to those structures do not contain sulfate, which may be useful for future experiments due to the presence of a sulfate ion in the active site of some of your structures. So it would be useful to state that the issues with collecting neutron diffraction data apply here because your particular structure belongs to the P6522 space group and that the resolution of your structure is 2.5 Å. This particular point should be mentioned somewhere in the results/discussion.

Response. The reviewer makes an interesting and important point. Structure 8AQL (unpublished) indeed has been obtained at high resolution and the unit cell dimensions point to the possibility to use similar crystallization conditions to grow neutron-quality crystals of hSHMT2. However in 8AQL, PLP is not in the internal aldimine state. It is rather a glycine-bound external aldimine. Nonetheless, we are pursuing similar complexes to obtain a neutron structure of hSHMT2. In case of 6DK3 (unpublished), it appears that there is no PLP present in the structure. Thus, this hSHMT2 structure is of the apo-form of the enzyme that is also dimeric instead of the usually tetrameric holoenzyme.

We have also stated in the second paragraph of page 8 that “We were unable to obtain neutron diffraction data from the P6522 crystals of hSHMT2 because they diffracted X-rays at a synchrotron only to 2.5 Å (Table S2).”

Minor points:

1. Page 5, line 141. Sentence should read ‘with the L-serine substrate’

Response. This change was applied to the text.

2. Page 9, line 241. Write full name (aspartate) instead of Asp.

Response. This change was applied to the text.

3. Page 9, line 250. Please write full enzyme name the first time the acronym AAT is used.

Response. This change was applied to the text.

4. Page 10, line 285. I would suggest a more speculative tone here as is used in the in the abstract. i.e. ‘suggesting the same protonation states’...

Response. The text was edited to read, “suggesting the same protonation states of O3’...”.

5. Page 11, line 305. ‘cryo-structure’ may make some readers think you are referring to a Cryo-EM structure. This needs to be rephrased so it is clear you mean crystals frozen in liquid nitrogen which had cryo-protectant added to them. I would also be good to point out somewhere in the manuscript that existing structures of hSHMT2 prior to this study all involved cryo-protected crystals, which is why your room temperature structure is particularly useful.

Response. We rephrased this sentence on page 11 to read “In contrast, a glycerol molecule is found in the substrate binding pocket in the X-ray structure of hSHMT2 obtained under cryogenic conditions with glycerol added as the cryoprotectant”. We also added the following sentence to the second paragraph on page 8: “Therefore, to directly compare hSHMT2 with the joint XN structure of *Tth*SHMT, we also determined a 2.5 Å room-temperature X-ray structure of hSHMT2 in the internal aldimine state, because all previous hSHMT2 X-ray structures were done at cryogenic temperatures.”

6. Page 11, line 320. ‘allowing us to conclude’ should be rephrased to something more speculative (see above).

Response. The text was edited to read, “allowing us to postulate that the His171 protonation state is the same...”

7. Page 11, line 327. What is the equivalent residue for Ser95?

Response. The equivalent for Ser95 in hSHMT2 is stated earlier in the same sentence.

8. Page 12, line 346. How conserved is the peripheral binding site (i.e. the folate binding site) between *Tth*SHMT and hSHMT2?

Response. The peripheral binding sites, where we observed L-Ser-d₇, in *Tth*SHMT and hSHMT2 are conserved. We added the equivalent residues for hSHMT2 on page 13.

9. Page 14, line 396. Have you tried collecting joint neutron/X-ray data for *Tth*SHMT/D-Ser?

Response. We could not collect neutron diffraction data on TthSHMT/D-Ser complex. We write at the top of page 14: “It was not however possible to produce neutron-quality crystals of this complex due to their size.” Please, also see our response to Reviewer 1 comment 1 on Experimental details.

10. Page 14, lines 409 and 410. Write full name (glutamate) instead of Glu.

Response. This change was applied to the text.

11. Other points:

12. *It would be useful to show a superposition of the overall TthSHMT dimer with hSHMT2.

Response. A supporting figure depicting the superposition of the overall TthSHMT and hSHMT2 dimers is shown as Figure S2.

13. *Please include a figure with your proposed mechanism for retro-aldol cleavage of L-serine into glycine.

Response. This figure has been included as panel **b** in Figure 5.

14. *Figure 5, it would be useful to depict where inhibitors/antifolates that are present in existing hSHMT2 structures bind relative to LLP and the ligands present in your structures.

Response. We attempted to generate such a figure by adding this information to the current Figure 5a. As there is already a lot of information pointing to the peripheral and cationic binding sites on this figure, it becomes not clear. We believe that the description of where usually inhibitors bind we provide in the Discussion is clear to relate to the figure. Therefore, we decided to keep Figure 5a as is.

REVIEWERS' COMMENTS:

Reviewer #1 (Remarks to the Author):

The authors have addressed all my points in a satisfactory way and I recommend it for publication in its present form.

Just one typo that I noticed. On page 20, "Hampton research" should be "Hampton Research".

Reviewer #2 (Remarks to the Author):

I agree with the author's responses.

Reviewer #3 (Remarks to the Author):

The authors have sufficiently addressed all of my points. The manuscript should be accepted for publication.